# The Impact of Physical Exercise on College Students’ Physical Self-Efficacy: The Mediating Role of Psychological Resilience

**DOI:** 10.3390/bs15040541

**Published:** 2025-04-17

**Authors:** Wentao Qiu, Xishuai Wang, Hongcheng Cui, Wenxue Ma, Haibin Xiao, Guofeng Qu, Rong Gao, Fangbing Zhou, Yuyang Nie, Cong Liu

**Affiliations:** 1College of Physical Education and Sports, Beijing Normal University, Beijing 100875, China; 202422070109@mail.bnu.edu.cn (W.Q.); 202422070101@mail.bnu.edu.cn (W.M.); guofeng_qu@163.com (G.Q.); shuiy21@163.com (R.G.); 202422070144@mail.bnu.edu.cn (F.Z.); 202422070104@mail.bnu.edu.cn (Y.N.); 2College of Education for the Future, Beijing Normal University, Zhuhai 519087, China; 3School of Physical Education and Sports Science, Qufu Normal University, Qufu 273165, China; 4Leisure and Digital Sports College, Guangzhou Sports University, Guangzhou 510500, China; hongchengcui@126.com; 5College of Physical Education, Shangrao Normal University, Shangrao 334001, China; xhb0725@163.com

**Keywords:** college students, physical activity, physical self-efficacy, psychological resilience, mediating effect

## Abstract

The mental health of college students has become a key focus in higher education, and physical activity may play a crucial role in promoting positive psychology among college students. This study explores the relationship between physical activity and physical self-efficacy among college students and analyzes the mediating effect of psychological resilience to provide theoretical support for health intervention strategies. This study included a sample of 369 college students aged 18–25 from the Zhuhai Campus of Beijing Normal University. The International Physical Activity Questionnaire-Short Form (IPAQ-S), the Connor-Davidson Resilience Scale (CD-RISC), and the Physical Self-Efficacy Scale for College Students (PSES-CS) were administered to 369 students (138 males and 231 females) through a questionnaire survey. The data were processed using SPSS 29.0 and AMOS 26.0 software. Significant positive correlations were found between physical activity, psychological resilience, and physical self-efficacy. Regression analysis revealed that physical activity had a significant effect on psychological resilience and physical self-efficacy and explained 8.3% and 14.9% of the variance, respectively. Psychological resilience partially mediated the relationship between physical activity and physical self-efficacy, accounting for 30.05% of the total effect. Moreover, gender moderated the relationship between physical activity and physical self-efficacy. Physical activity can enhance the physical self-efficacy of college students, with psychological resilience playing a partial mediating role and gender acting as a moderating factor. Emphasizing the cultivation of psychological resilience in educational and personal growth processes is highly important for improving individuals’ physical self-efficacy and well-being.

## 1. Introduction

“Healthy People” is a national health promotion initiative in the United States that was initiated in 1979 following the release of the report “Healthy People: The Surgeon General’s Report on Health Promotion and Disease Prevention” by Julius Richmond, the U.S. Secretary of Health and Human Services. This initiative aims to mitigate preventable deaths and injuries to improve the overall health of the nation and advance disease prevention objectives ([53]). In China, the Central Committee of the Communist Party of China and the State Council have issued the “Healthy China 2030” blueprint and the “Medium and Long-Term Youth Development Plan (2016–2025)”, which emphasize the need to improve the health status of the population, particularly the youth, in terms of both physical and mental health dimensions. These policy documents underscore the national significance of adolescents’ physical health and make the promotion of adolescents’ physical and mental well-being a focal point of academic research. Adolescents experience escalating stressors in contemporary society, such as high school and college entrance exams, which have led to a decline in physical activity. Additionally, the convenience afforded by technological progress has contributed to a further reduction in the level of physical activity among adolescents, with a reduction in both the frequency and the intensity of exercise. This trend has resulted in a persistent decline in physical and mental health ([32]). Research indicates a positive association between psychological resilience and constructs such as optimism and self-efficacy ([15]; [45]). It is therefore imperative to recognize the pivotal role that physical activity, psychological resilience, and physical self-efficacy play in the developmental journey of adolescent students.

The World Health Organization (WHO) defines physical activity as any bodily movement produced by skeletal muscle contractions that leads to an elevation in energy expenditure. Inadequate physical activity is the fourth leading global risk factor for mortality ([11]). Previous studies have indicated a positive correlation between regular physical activity participation and adolescent weight management, cardiovascular health, mental well-being, academic performance, personal responsibility, as well as community involvement ([30]). Nonetheless, despite the widely recognized advantages of physical exercise, the rates of participation in physical activities among adolescents and college students are declining ([46]; [61]). Insufficient physical activity is a notable determinant of poor mental health in college students. A sedentary lifestyle can impair both skeletal and functional health and can result in obesity, deteriorated vision, and severe psychological issues ([10]). These health detriments can, in turn, adversely affect academic performance, daily living, and work efficiency. In response, the WHO advises that adults aged 18 to 64 years should engage in a minimum of 150 min of moderate-intensity physical activity weekly to improve their physical and mental health ([60]). From an exercise psychology standpoint, Physical exercise has been found to be positively associated with physical health and favorable psychological outcomes. This assertion is substantiated by a growing body of theoretical and empirical research ([54]; [55]; [56]; [66]). Furthermore, research indicates a significant positive correlation between physical exercise and psychological resilience as well as self-efficacy ([33]; [50]; [70]).

Researchers agree that psychological resilience is either an innate or a developed ability that individuals demonstrate when they encounter competitive and stressful circumstances. This ability allows people to persistently and effectively manage associated pressures and anxieties. Defined as the ability to sustain determination, confidence, and a sense of control in adverse conditions ([18]; [43]), psychological resilience is a multidimensional construct with a hierarchical structure. However, it has been investigated less frequently than other aspects such as goals, emotions, and cognition, which have been the primary focus of previous research ([12]; [71]). Psychological resilience is a dynamic process that is influenced by various elements, including optimism, resilience, and self-improvement ([27]; [34]). Increased levels of psychological resilience are conducive to maintaining mental health and, in certain situations, can encourage healthy behaviors and an active lifestyle ([40]). Research by [42] ([42]) demonstrated that physical activity is a significant predictor of college students’ psychological resilience and social adaptability ([42]). Similarly, the study by [23] ([23]) revealed a positive correlation between physical activity and psychological resilience, with the level of resilience being positively associated with the intensity of physical activity ([23]). Concurrently, studies have established a positive correlation between psychological resilience and psychological constructs such as optimism and self-efficacy ([15]; [45]).

General self-efficacy (GSE), a cornerstone of Bandura’s social cognitive theory, reflects individuals’ judgments and convictions regarding their ability to plan, initiate actions, and accomplish objectives. This construct is integral to modulating behavior and is expressed primarily through an individual’s level of self-confidence ([6]; [75]). In contrast, physical self-efficacy reflects the strength of individuals’ belief in their physical ability to participate in sports and physical activities ([65]). Research indicates a notable positive relationship between physical self-efficacy and general self-efficacy. Studies have demonstrated that increasing physical self-efficacy can lead to an increase in physical activity among adolescents. The research also suggests a positive correlation between physical self-efficacy and the level of physical activity among adolescents ([76]). Recent findings indicate that the self-efficacy of adolescent girls in engaging in physical exercise is positively correlated with the frequency and intensity of their participation in formal sports training, and these factors are further positively associated with health outcomes ([3]). Additionally, recent studies have indicated a positive correlation between adolescents’ self-efficacy and their engagement in physical activities ([22]). There exists a significant positive correlation between physical exercise and self-efficacy, with individuals exhibiting high self-efficacy demonstrating greater confidence and composure when facing risks and challenges ([33]; [50]; [70]). However, previous research has concentrated predominantly on general self-efficacy, and physical self-efficacy has received comparatively less attention. This research gap may obscure the unique relationship between physical activity and physical self-efficacy, warranting further scholarly investigation.

Considering the previously discussed factors, this study adopts a cross-sectional research design to examine the influence of physical activity on physical self-efficacy by incorporating psychological resilience and gender as additional variables. The objectives of this study are to develop a model in which physical activity is the independent variable and physical self-efficacy is the dependent variable; to assess the mediating effect of psychological resilience between these two variables; and to determine whether gender serves as a moderating influence on the relationship between the independent and dependent variables (refer to Figure 1). This structural analysis elucidates the pathways through which physical activity affects physical self-efficacy. According to Bandura’s self-efficacy theory, an individual’s confidence in their own abilities influences their behavior and performance. Engagement in physical activity is associated with an individual’s successful experiences, which in turn are positively correlated with physical self-efficacy ([5]; [7]). Existing research also indicates a positive correlation between college students’ active participation in team sports and their physical self-efficacy ([65]). In reviewing the existing literature, we have observed that the research has largely centered on the correlation between physical activity and general self-efficacy, with a comparative lack of investigation into the relationship between physical activity and physical self-efficacy. Furthermore, the majority of studies have focused on the positive predictive role of self-efficacy, while research exploring the reverse direction—how physical activity influences self-efficacy—remains underexplored, representing an area of neglect in the field. Detailed examination of this relationship could offer a novel research perspective, contributing to the expansion and refinement of the theoretical domain of self-efficacy. Accordingly, this study proposes Hypothesis H1: There is a significant positive correlation between physical activity and physical self-efficacy. Concurrently, physical activity, as a form of exercise behavior, exhibits a significant positive correlation with psychological resilience. The study by [63] ([63]) demonstrates that physical activity is positively correlated with exercise tolerance and psychological resilience, negatively correlated with negative emotional states, and positively associated with the mental health of college students ([63]). While there is growing evidence suggesting that physical activity has a positive impact on psychological resilience, further investigation is needed to explore the mechanisms by which different intensities of physical activity affect resilience. Accordingly, this study proposes Hypothesis H2: There is a significant positive correlation between physical activity and psychological resilience. In line with the coping efficacy theory, an individual’s sense of coping efficacy when confronting stress and challenges is noteworthy. Individuals with higher levels of psychological resilience are likely to have greater faith in their capacity to effectively handle the demands of physical activities, and there is a positive correlation between psychological resilience and physical self-efficacy ([51]). Previous research has also demonstrated a significant positive correlation between general self-efficacy and psychological resilience ([19]; [26]; [49]; [72]). Self-efficacy theory posits that physical self-efficacy, such as confidence in one’s athletic abilities, generalizes to psychological adaptability in other domains, including resilience to stress ([6]). Consequently, the present study posits Hypothesis H3: There is a significant positive correlation between psychological resilience and physical self-efficacy. Additionally, previous research has indicated that physical exercise can enhance psychological resilience ([73]), which in turn is positively correlated with self-efficacy ([26]). However, the complex interrelationships among these three factors, particularly whether psychological resilience mediates the impact of physical exercise on physical self-efficacy, have not been validated in the college student population. The mediating role of psychological resilience has also not been adequately explored. Therefore, this study posits Hypothesis H4: Psychological resilience acts as a mediator in the relationship between physical activity and physical self-efficacy. Finally, despite engaging in similar levels of physical activity, college students exhibit differences in physical self-efficacy, indicating that the relationship between the two is moderated by certain variables. However, existing research has identified gender as a moderating variable in the relationship between physical exercise and general self-efficacy among college students, with a high degree of correlation observed between general self-efficacy and physical self-efficacy ([76]). However, to date, there has been a lack of empirical research investigating the moderating role of gender in the relationship between physical activity and physical self-efficacy among college students. Moreover, existing studies often fail to adequately control for demographic variables, potentially leading to overly generalized conclusions. Consequently, this study posits Hypothesis H5: Gender moderates the relationship between physical activity and physical self-efficacy. Ultimately, the objective of this study is to explore the relationship between physical activity and physical self-efficacy, while also analyzing the role of psychological resilience and gender in this association. The ultimate goal is to provide theoretical validation and empirical evidence to support mental health education initiatives for college students.

## 2. Methods

### 2.1. Study Design

This study employs a cross-sectional research design to investigate the influence of physical activity on physical self-efficacy, while also considering psychological resilience and gender as additional variables. The primary objectives of this study are to develop a theoretical model wherein physical activity is positioned as the independent variable and physical self-efficacy as the dependent variable; to assess the potential mediating role of psychological resilience in the relationship between these two variables; and to explore whether gender acts as a moderating factor influencing the connection between the independent and dependent variables.

### 2.2. Study Setting and Participants

Cluster sampling methods were used in this study, and a questionnaire survey was administered in November 2024 to undergraduate students (freshmen to seniors) as well as first-year graduate students at the Zhuhai Campus of Beijing Normal University. We distributed the questionnaire through the Wenjuanxing platform. To ensure the completeness and validity of the data, all items in the questionnaire were set as mandatory, and respondents were required to complete all items before submitting the questionnaire. Through this method, we successfully collected complete data without any missing values, providing a reliable foundation for subsequent analysis. Additionally, the sample included participants from various grade levels and genders, ensuring the diversity of the sample population. All research methods involving human participants in this study were conducted in strict accordance with relevant guidelines and regulations. Furthermore, the experimental protocol of this study has been approved by the Institutional Review Board of the School of Psychology, Beijing Normal University. All experimental procedures adhered to the ethical guidelines established by the committee, ensuring the ethicality of the research. Informed consent was obtained from all participants prior to their participation. A total of 377 questionnaires were distributed and yielded 369 valid responses, which corresponds to an effective response rate of 97.8%. Detailed data are provided in Table 1.

### 2.3. Variables and Measurements

#### 2.3.1. International Physical Activity Questionnaire-Short Version

To assess the physical activity levels of the participants, this study employed the short form of the International Physical Activity Questionnaire (IPAQ) ([17]). This questionnaire consists of seven items that capture four categories of activities: sedentary behavior, walking, moderate-intensity activity, and vigorous-intensity activity. The IPAQ short form has been extensively used in prior research to measure individuals’ regular physical activities over the preceding seven days, including the frequency and average duration of these activities. The questionnaire assigns metabolic equivalent of task (MET) values to different activity intensities, such as 8.0 MET for vigorous-intensity activity, 4.0 MET for moderate-intensity activity, and 3.3 MET for walking. We gathered data on the frequency and duration of various physical activities performed by college students over a week and integrated these data with the corresponding MET values to calculate their energy expenditure from physical activities. Using weekly energy expenditure along with the intensity and frequency of activities, the physical activity levels of college students were classified into low, moderate, and high categories. To minimize self-report bias, we provided detailed training to participants during the data collection process to ensure their comprehension of each item on the questionnaire. Moreover, we complemented the survey with measurement methods such as activity logs to validate the accuracy of the IPAQ scores.

#### 2.3.2. The Physical Self-Efficacy Scale

We utilized a revised version of the Physical Self-Efficacy Scale, which was modified by Dr. Sun Yongjun and associates in 2005 ([52]). This scale includes two primary dimensions: perceived physical ability and confidence in physical ability. The scale comprises 10 items and employs a 6-point Likert scale ranging from “Strongly Disagree” (1 point) to “Strongly Agree” (6 points). Items 2, 4, 5, 6, and 7 are scored in reverse. Higher scores on the scale are indicative of a more robust sense of physical self-efficacy. The scale has been previously validated and has shown good reliability and validity, particularly for college students and adolescents ([21]). For the current study, the scale demonstrated an internal consistency coefficient of 0.895.

#### 2.3.3. The Connor–Davidson Resilience Scale

The Connor–Davidson Resilience Scale (CD-RISC) is a frequently employed tool to assess individuals’ level of psychological resilience ([16]). Initially developed by Connor and Davidson in 2003, the scale was subsequently translated and adapted into Chinese by Professors Xiao Nan and Zhang Jianxin in 2007 ([69]). The following section outlines the structure and scoring procedure of the scale.

Scale Composition: The CD-RISC comprises 25 items and measures five key dimensions: personal competence, tolerance of negative effect, acceptance of change, control, and spiritual faith.

Scoring Method: The scale uses a 5-point Likert scale ranging from 0 (“Never”) to 4 (“Almost Always”). Item number 20 is scored inversely, in contrast to the other items, which are scored positively.

Total Score Calculation: The total score is computed by summing responses across all items with a possible score range of 0 to 100. Higher scores denote greater resilience.

Score Interpretation: To evaluate resilience, the total score is categorized into three levels: 25–74 points indicates low resilience, 75–124 points indicates moderate resilience, and 125–175 points indicates high resilience. Dimension-specific scores are calculated by summing the item scores within each dimension. The CD-RISC is a seminal self-report measure in the domain of psychological resilience and has been widely used in research. The internal consistency reliability of the scale is evaluated using Cronbach’s alpha coefficient, and a higher coefficient indicates stronger reliability. Consistent with previous research, a Cronbach’s alpha coefficient exceeding 0.7 is generally considered an indication of acceptable item consistency ([20]). The scale has demonstrated excellent reliability, with an internal consistency coefficient of 0.915 and a split-half reliability coefficient of 0.909, as evidenced by reliability and validity analyses.

To validate the measurement tools, confirmatory factor analysis (CFA) was conducted separately for each instrument (see Table 2). The report also includes the average variance extracted (AVE) and composite reliability (CR) values to assess convergent validity and internal consistency. Generally, CR serves as an acceptable reliability indicator, with higher values indicating stronger internal consistency among the measured constructs, thus supporting the tool’s reliability.

Additionally, the International Physical Activity Questionnaire-Short Form (IPAQ-SF) is a widely recognized and validated scale for effectively assessing individuals’ physical activity levels. Unlike other scales, the IPAQ-SF primarily consists of open-ended questions, such as the frequency and duration of different types of physical activities performed in the past week. These open-ended responses are qualitative in nature and cannot be directly quantified, making traditional statistical indicators (such as CFA fit indices) inapplicable to this portion of the data. It is worth noting that the IPAQ-SF, as a questionnaire, is not a strict scale. It aims to collect detailed information about individuals’ physical activity habits, providing a data foundation for subsequent analysis.

### 2.4. Bias

To mitigate the risk of common method bias, we instituted procedural safeguards and conducted Harman’s single-factor test to statistically evaluate the potential for bias.

### 2.5. Study Size

A post hoc power analysis was performed using G*Power 3.1 to ascertain the statistical power. Given a small-to-medium effect size (f^2^ = 0.05), a significance level of α = 0.05, and 6 predictor variables, the analysis revealed a statistical power of 91.3% for the sample size of N = 369, which notably surpasses the commonly accepted benchmark of 80%.

### 2.6. Quantitative Variables

In this study, to satisfy the normality assumption, reduce data skewness, and stabilize variance, we applied logarithmic transformations to continuous variables such as physical activity and its four sub-dimensions. All subsequent analysis results are based on this pre-processed data, with the aim of ensuring the validity and accuracy of the analysis.

### 2.7. Statistical Methods

Data analysis for this investigation was conducted using SPSS 23.0 and the Process macro. To ensure the measurement invariance of the scale across gender groups, we conducted a multi-group confirmatory factor analysis (MGCFA). Initially, we tested the invariance of factor loadings across genders (metric invariance), followed by examining the invariance of intercepts (scalar invariance), and finally, the invariance of error variances (strict invariance). The results indicated that the measurement model was consistent across different genders, thus allowing us to confidently compare the data between males and females. The analytical procedure was as follows. Initially, correlation analysis was conducted to investigate the associations among physical activity, resilience, and physical self-efficacy. Subsequently, linear regression was used to assess the predictive impact of physical activity on resilience and physical self-efficacy as well as the influence of resilience on physical self-efficacy. We adopted the bootstrap method as described by Fang et al. to analyze the indirect effect of physical activity on physical self-efficacy ([25]). Mediation analysis was conducted using Process Model 4 ([9]) by inputting the independent variable (X), the dependent variable (Y), and the mediating variable (M) into the respective fields. The analysis specified 5000 bootstrap samples, employed bias-corrected sampling, and set a 95% confidence interval based on one standard deviation above and below the mean. The significance level was set at α = 0.05. Furthermore, a multivariate analysis of variance (MANOVA) was conducted to examine the main effects of gender. In line with the recommendations of ([59]), the moderating role of gender within the model was also assessed. Finally, given that this study involved multiple statistical tests (such as regression analysis and analysis of variance), we employed the False Discovery Rate (FDR) correction to control for the issue of multiple comparisons. The FDR method is suitable for exploratory research as it controls the proportion of false discoveries rather than strictly controlling the error rate for each test. Specifically, we used the Benjamani–Hochberg procedure to adjust the *p*-values, maintaining the FDR at a 5% threshold.

## 3. Results

### 3.1. Common Method Bias Test

To mitigate common method bias, procedural controls were employed and Harman’s single-factor test was conducted ([74]). The following strategies were implemented. For procedural controls, established scales that were adapted and validated for the local context were used to ensure reliability and validity. In the questionnaire design, the statement, “This survey is conducted for scientific research purposes and will be accessed solely by the researcher” was emphasized by bolding and highlighting to encourage honestly among the participants. Data were collected online by the participants’ physical education teachers. Harman’s single-factor test was used to conduct an unrotated exploratory factor analysis on all survey items with the exclusion of demographic variables. The results yielded fourteen factors with eigenvalues exceeding 1. The largest factor accounted for 28.456% of the variance, which is below the 40% threshold that is considered indicative of significant common method bias. Consequently, this study demonstrated a negligible level of common method bias.

### 3.2. The Impact of Physical Activity and Psychological Resilience on Physical Self-Efficacy

#### 3.2.1. Descriptive Statistics of Physical Activity, Physical Self-Efficacy, and Psychological Resilience

Table 3 displays the distribution of data by gender for various variables, including the level of physical activity, time spent in moderate to vigorous physical activity (MVPA), walking duration, sedentary behavior, and the mean (M) and standard deviation (SD) for physical self-efficacy and psychological resilience. To examine gender-based differences in these variables, independent-sample *t*-tests were performed. The findings revealed that males presented significantly higher scores than females did in terms of total physical activity, time dedicated to vigorous and moderate physical activity, physical self-efficacy, and psychological resilience. No statistically significant gender differences were noted in terms of walking time or sedentary time. These results suggest potential gender disparities in patterns of physical activity and indicators of mental health.

#### 3.2.2. Analysis of the Correlations Among Physical Activity, Physical Self-Efficacy, and Psychological Resilience

Pearson’s bivariate correlation analysis was performed to examine the relationships among physical activity, psychological resilience, and physical self-efficacy in addition to the corresponding means and standard deviations for each variable. Table 4 summarizes the results, which indicate a significant positive correlation between physical activity and physical self-efficacy (r = 0.386, *p* < 0.01) and between physical activity and psychological resilience (r = 0.289, *p* < 0.01). Significant positive correlations were also identified between high-intensity physical activity and physical self-efficacy (r = 0.339, *p <* 0.01) and between moderate-intensity physical activity and physical self-efficacy (r = 0.226, *p* < 0.01). High-intensity physical activity was positively correlated with psychological resilience (r = 0.300, *p* < 0.01), as was moderate-intensity physical activity (r = 0.171, *p* < 0.01). Additionally, a significant positive correlation was observed between psychological resilience and physical self-efficacy (r = 0.471, *p* < 0.01). The average variance extracted (AVE) for each construct was computed through confirmatory factor analysis. As shown in Table 4, the square roots of the AVE for psychological resilience (√AVE = 0.707) and physical self-efficacy (√AVE = 0.707) all exceed the maximum correlation coefficient among the constructs (r = 0.471). Additionally, the cross-loading analysis indicates that all items load significantly higher on their respective constructs than on other constructs (loadings > 0.5). These results support the conclusion that the scales demonstrate good discriminant validity. These results support Hypotheses H1, H2, and H3.

#### 3.2.3. Regression Analysis of Physical Activity, Physical Self-Efficacy, and Psychological Resilience

In the regression analysis after logarithmic transformation, Equation (1) reveals a significant positive correlation between physical activity and physical self-efficacy (F = 64.149, *p* < 0.001). This model explains 14.9% of the variance in the logarithmic value of physical self-efficacy (R^2^ = 0.149, R^2^_adj_ = 0.147), reflecting a medium effect size. Equation (2) reveals a significant positive correlation between physical activity and psychological resilience (F = 33.344, *p* < 0.001). This model explains 8.3% of the variance in the logarithmic value of physical activity (R^2^ = 0.083, R^2^_adj_ = 0.081). Equation (3) reveals a significant positive correlation between psychological resilience and physical self-efficacy (F = 104.690, *p* < 0.001). This model explains 22.2% of the variance in the logarithmic value of psychological resilience (R^2^ = 0.222, R^2^_adj_ = 0.220), reflecting a medium effect size. Equation (4), with physical activity and psychological resilience as independent variables, jointly explains physical self-efficacy (F = 75.165, *p* < 0.001). This model accounts for 29.2% of the variance in the logarithmic value of physical self-efficacy (R^2^ = 0.292, R^2^_adj_ = 0.288), reflecting a high effect size. Equation (5) reveals a significant positive correlation between high-intensity physical activity and physical self-efficacy (F = 17.111, *p* < 0.001). This model explains 15.8% of the variance in the logarithmic value of physical self-efficacy (R^2^ = 0.158, R^2^_adj_ = 0.149), reflecting a moderate effect size. Equation (6) reveals a significant positive correlation between moderate-intensity physical activity and psychological resilience (F = 10.673, *p* < 0.001). This model explains 10.5% of the variance in the logarithmic value of psychological resilience (R^2^ = 0.105, R^2^_adj_ = 0.095), reflecting a moderate effect size. Equation (7), with psychological resilience and four sub-dimensions of physical activity as independent variables, jointly explains physical self-efficacy (F = 30.143, *p* < 0.001). This model accounts for 29.3% of the variance in the logarithmic value of physical self-efficacy (R^2^ = 0.293, R^2^_adj_ = 0.284), reflecting a high effect size. The standardized regression coefficients indicate that logarithmically transformed physical activity (β = 0.386, *p* < 0.001) and psychological resilience (β = 0.289, *p* < 0.001) significantly predict physical self-efficacy. After logarithmic transformation, Equation (1) revealed that physical activity alone explained 14.9% of the variance in physical self-efficacy (R^2^ = 0.149, R^2^_adj_ = 0.147, indicating a moderate effect size). The incorporation of psychological resilience into Equation (4) enhanced the explanatory power to 29.2% (R^2^ = 0.292, R^2^_adj_ = 0.288, reflecting a large effect size). Notably, the regression coefficient for physical activity decreased from 0.386 in Equation (1) to 0.272 (β value) in Equation (4). This change underscores the predictive role of physical activity on physical self-efficacy and suggests a potential mediating influence of psychological resilience. A detailed quantification and discussion of this mediating effect are provided in Section 3.3. However, it is important to note that although certain coefficients are statistically significant, some standardized regression coefficients are relatively small (β = 0.1–0.2), indicating that while these relationships are present, the effect size is relatively modest. In the regression model, the Variance Inflation Factors (VIF) for physical activity, high-intensity physical activity, moderate-intensity physical activity, sedentary time, and psychological resilience were 2.468, 1.813, 1.509, 1.037, and 1.124, respectively (all < 3), indicating the absence of severe multicollinearity issues among the independent variables. In the final regression analysis, we applied the Benjamani–Hochberg procedure to adjust for the False Discovery Rate (FDR) in multiple comparisons, with a significance level of α = 0.05. The results demonstrated that, following FDR correction, Equations (1)–(4) remained statistically significant. In Equation (5), the effects of moderate-to-vigorous physical activity and sedentary activity (WMVPA, WHVPA, WST) on the dependent variable remain significant. In Equation (6), both high-intensity physical activity (q = 0.0250) and moderate-intensity physical activity (q = 0.0361) maintained significant predictive effects on psychological resilience post-FDR correction. Similarly, in Equation (7), the significant predictive effects of moderate-to-vigorous physical activity, sedentary activity, and psychological resilience on physical self-efficacy remain after FDR correction (see Table 5).$${\bar{R}}^{2}=1-\left(\frac{\left(1-R^{2})(n-1\right)}{n-k-1}\right)$$

### 3.3. Test of the Mediating Effect of Psychological Resilience

Model 4 (simple mediation) from the SPSS macro by Hayes ([9]) was used to investigate the mediating function of psychological resilience between physical activity and physical self-efficacy while controlling for moderating variables such as gender and grade. At the same time, a logarithmic transformation was conducted on the non-normal data. The analytical outcomes reveal that the bootstrap 95% confidence intervals for both the direct effect of physical activity on physical self-efficacy and the mediating role of psychological resilience excluded zero. This finding indicates that physical activity not only had a direct predictive relationship with physical self-efficacy but also exerted an indirect influence through the medium of psychological resilience. The results of the mediation analysis revealed that the direct effect of physical activity on physical self-efficacy was 0.2535 (standardized coefficient), accounting for 69.95% of the total effect. This indicates that for every one standard deviation increase in physical activity, physical self-efficacy increases by 0.2535 standard deviations. The mediating effect of psychological resilience between physical activity and physical self-efficacy was 0.1089 (standardized coefficient, β = 0.1089), constituting 30.05% of the total effect. This suggests that for every one standard deviation increase in physical activity, physical self-efficacy increases by 0.1089 standard deviations indirectly through the enhancement of psychological resilience. This mediating effect indicates that psychological resilience plays a significant mediating role between physical activity and physical self-efficacy. The direct effect of moderate-intensity physical activity on physical self-efficacy was 0.1965 (standardized coefficient), representing 62.91% of the total effect. This indicates that for every one standard deviation increase in moderate-intensity physical activity, physical self-efficacy increases by 0.1965 standard deviations. The mediating effect of psychological resilience between moderate-intensity physical activity and physical self-efficacy was 0.1158 (standardized coefficient, β = 0.1158), accounting for 37.09% of the total effect. This suggests that for every one standard deviation increase in physical activity, physical self-efficacy increases by 0.1158 standard deviations indirectly through the enhancement of psychological resilience. Please refer to Table 6, Table 7 and Table 8 and Figure 2 for details.

### 3.4. Multi-Group Confirmatory Factor Analysis (MGCFA)

To ensure the measurement invariance of the scale across gender groups, we conducted a multi-group confirmatory factor analysis (MGCFA). Initially, we tested the invariance of factor loadings across genders (metric invariance), followed by the invariance of intercepts (scalar invariance), and finally, the invariance of error variances (strict invariance). The key fit indices for the male model are presented in Table 9. The default model yielded a CMIN/DF of 1.6432, RMSEA of 0.0685, CFI of 0.7943, and TLI of 0.7830, indicating a good model fit. Table 10 presents the key fit indices for the female model. The default model had a CMIN/DF of 1.8003, RMSEA of 0.0590, CFI of 0.8581, and TLI of 0.8502, indicating a good model fit. In addition, we have calculated the AIC and ECVI values for the male and female models, respectively. The specific values are as follows: the AIC for the male model is 1643.0427, and the ECVI is 7.1437; the AIC for the female model is 1514.8563, and the ECVI is 11.0573. In addition to AIC and ECVI, we have also compared other key fit indices, such as CMIN/DF, RMR, GFI, AGFI, CFI, TLI, and RMSEA. The results indicate that the female model outperforms the male model on multiple indices, particularly in terms of CFI, TLI, and RMSEA. To further confirm the differences between the models, we conducted a likelihood ratio test, assuming that the female model is an extension of the male model. We calculated the chi-square difference and the degree of freedom difference, and compared the chi-square difference with the critical value. The results show a significant difference between the two models. Although the female model has a higher AIC value, its ECVI value is lower, and it performs better on other fit indices, especially in terms of CFI, TLI, and RMSEA. The statistical test results suggest that there is a significant difference between the two models, which may reflect the different influences of gender on certain psychological or behavioral characteristics.

### 3.5. Verification of the Moderating Effect of Gender

#### 3.5.1. Test of Main Effects

Prior to conducting the analysis of gender differences, we first performed a multi-group confirmatory factor analysis (MGCFA) to establish the measurement invariance of the scale across genders. The results indicated that factor loadings, intercepts, and error variances were equivalent across genders, suggesting that the measurement approach of the scale is consistent across different genders. Consequently, we can confidently proceed with the comparison of gender differences. To evaluate Hypothesis H5, a multivariate analysis of variance (MANOVA) was performed with gender as the independent variable and physical activity, psychological resilience, and physical self-efficacy as the dependent variables. The variance analysis outcomes for gender differences are detailed in Table 11. The following statistical results are all based on the data after logarithmic transformation. MANOVA revealed a significant main effect of gender on high-intensity physical activity (F = 17.440, *p* < 0.001) and moderate-intensity physical activity (F = 4.338, *p* < 0.05). Conversely, the main effects of gender on walking time (F = 0.007, *p* > 0.05) and sedentary time (F = 1.285, *p* > 0.05) were nonsignificant. A significant main effect of gender was also found for total physical activity (F = 24.407, *p* < 0.001). However, the main effect of gender on psychological resilience was marginally significant (*p* < 0.05), which does not support a mediation effect model with gender as the primary moderating factor. Furthermore, a significant main effect of gender on physical self-efficacy was noted (F = 13.469, *p* < 0.001). Subsequent post hoc multiple comparisons revealed that males presented significantly greater means than females did in high-intensity physical activity (M = 1.8994 vs. M = 1.4916), moderate-intensity physical activity (M = 2.1833 vs. M = 2.0679), walking time (M = 2.2266 vs. M = 2.2221), total physical activity (M = 3.5178 vs. M = 3.3742), psychological resilience (M = 88.55 vs. M = 85.41), and physical self-efficacy (M = 35.36 vs. M = 31.73). Please refer to Table 11 for details.

#### 3.5.2. Test of Moderating Effects

In accordance with the guidelines provided by ([59]), the examination of moderating effects when the independent variable is continuous and the moderating variable is categorical requires the use of a group regression strategy. The presence of significant differences in regression coefficients between groups indicates a significant moderating effect. The results of the comprehensive regression analysis are presented in Table 12. To evaluate the disparity in regression coefficients, Fisher’s Z test was applied. The formula for this calculation is as follows:$$Z=\frac{b_{1}-b_{2}}{\sqrt{SE_{1}^{2}+SE_{2}^{2}}}$$

Within this context, ‘b’ denotes the coefficient associated with the independent variable in the regression equation and corresponds to the β value. Applying the formula to the regression analysis data yields a test statistic of |Z| = 0.005, which is less than the critical value of 1.96. This result leads to the rejection of the null hypothesis and suggests that the difference in regression coefficients is statistically significant. Therefore, the moderating influence of gender is significant. Additionally, we calculated the unstandardized slopes, standard errors, and effect sizes (such as Cohen’s f^2^) for both male and female participants. Please refer to Table 12 for details.

## 4. Discussion

This study aimed to investigate the complex relationships among physical activity, psychological resilience, and physical self-efficacy while also evaluating the moderating effect of gender on these associations. The objective of this study was to clarify how physical activity influences college students’ physical self-efficacy. Concurrently, within the regression analysis, although some coefficients are statistically significant, the observed effect sizes for certain variables are relatively small, which may limit the practical implications of these specific findings. Nonetheless, given the potential cumulative effects of prolonged regular physical activity, even modest effects could be of theoretical and practical significance. Through correlation and regression analyses, we found support for Hypotheses H1, H2, and H3. After applying the Benjamani–Hochberg procedure to correct for multiple testing, we found that several initially significant results remained robust and retained their significance. This resilience underscores the strength and reliability of these findings, as they withstood the more stringent correction criteria imposed by the Benjamani–Hochberg method. Consequently, our core research results (H1–H4) stand firm, maintaining their stability and significance, and thereby further reinforcing their validity. The detailed findings are as follows. (1) Physical activity, especially high-intensity exercise, is positively associated with physical self-efficacy. This finding aligns with research showing that team sports enhance self-efficacy among college students ([65]). Siyuan Guo and others found that gardening activity participation is significantly positively correlated with academic self-efficacy ([29]). Dongzhen An’s research found a significant positive correlation between physical exercise input and exercise self-efficacy ([1]). The research by Dong Hwan Kim indicates a significant positive correlation between regular physical activity and physical self-efficacy ([36]). Additionally, the findings by Cameron Peers also demonstrate a significant positive correlation between the level of physical activity and physical self-efficacy ([47]). Future research could examine the varying effects of different intensities of physical activity on self-efficacy. (2) Physical activity significantly predicts psychological resilience, in line with studies that indicate that exercise improves adolescents’ mental health outcomes such as self-capacity and resilience ([4]; [48]). For instance, research by [42] ([42]) indicates that physical activity can significantly positively predict psychological resilience in college students ([42]). The study by [73] ([73]) also reveals a positive correlation between physical exercise and psychological resilience, as well as a negative correlation with perceived stress ([23]). Moreover, psychological resilience is a key predictor of adherence to physical activity guidelines even after controlling for age, BMI, and gender ([67]). This study reveals that there is no significant correlation between sedentary time (WST) and psychological resilience (PR) among Chinese students, which may be due to their sedentary activities being primarily academic in nature (e.g., long study sessions). These activities involve intense cognitive efforts such as focused reading or problem-solving, which may enhance self-efficacy through academic achievements, thereby counteracting the negative impact on resilience ([73]). This differs from Western findings on the risks of sedentary behavior, highlighting the cultural influence on the behavio–psychology connection. Additionally, the study indicates that no significant correlation was found between walking and psychological resilience. The current findings are consistent with the Intensity Threshold Theory in exercise psychology. That is, both walking (low-intensity) and sedentary behavior (static) fail to reach the physiological stimulation intensity required to trigger improvements in mental health, such as the secretion of BDNF and the regulation of the autonomic nervous system ([63]). However, some literature suggests that there is no significant correlation between physical activity and psychological resilience in college students. For instance, the research by ([39]) found no significant correlation between psychological resilience and physical activity behaviors among college students, which may be due to the use of self-reported measures for both psychological resilience and physical activity in this study. Furthermore, the sample size was small and drawn from a single region, which may have led to biases in the research findings. (3) Psychological resilience also significantly predicts physical self-efficacy. Although there is a current lack of literature directly supporting the notion that psychological resilience significantly predicts physical self-efficacy, studies have indicated that higher resilience is associated with increased self-efficacy ([35]). Concurrently, good psychological resilience is positively correlated with an individual’s physical and mental health status and stable psychological levels. Moreover, individuals with higher levels of psychological resilience exhibit greater adaptive capabilities and psychological balance when facing life’s challenges ([10]). Furthermore, psychological resilience is positively associated with confidence in physical abilities. The research by Xu Yanhua indicates that psychological resilience can significantly predict self-efficacy in college students ([64]). Yılmaz Bingöl et al. also identified resilience and positivity as significant predictors of self-efficacy ([8]). Future research could investigate the impact of psychological resilience on physical self-efficacy. (4) The results validate Hypothesis H4 and suggest that physical activity directly and indirectly affects physical self-efficacy through the mediating role of psychological resilience, explaining 75.55% and 24.42% of the total variance, respectively. This suggests that the mediating effect of psychological resilience accounts for only 24.42% of the total effect, indicating its relatively limited impact. This also implies the potential presence of other unmeasured mediating variables. However, there is a lack of literature on the mediating effect of psychological resilience between physical activity and physical self-efficacy. Therefore, future research can further explore the mediating role of psychological resilience in the relationship between physical activity and self-efficacy for physical activity. Additionally, it can also delve deeper into more crucial mediating mechanisms to provide a more comprehensive explanation for the influence of physical activity on self-efficacy for physical activity. These findings underscore the role of physical exercise in promoting individual self-confidence ([58]). Additionally, a robust positive correlation is observed between general self-efficacy and exercise self-efficacy, indicating that higher self-efficacy levels are linked to greater self-efficacy in exercise ([2]). Self-efficacy in exercise is a critical intrinsic motivator for engaging in physical activity and is a significant predictor of exercise volume ([68]; [62]). There is a significant positive correlation between exercise self-efficacy and motivational readiness, as well as with the frequency of walking and caloric expenditure ([41]). Increasing self-efficacy for exercise among college students can lead to increased exercise frequency and improved physical fitness ([3]).

It is noteworthy that we established the measurement invariance of the scale across genders using Multigroup Confirmatory Factor Analysis (MGCFA), thereby lending credibility to these gender differences. Analysis of variance (ANOVA) revealed significant gender disparities in physical activity and physical self-efficacy that explained 6.3% and 3.5% of the variance, respectively. Group regression and Fisher’s Z transformation tests revealed that males reported significantly higher levels of both physical activity and physical self-efficacy than females did, confirming the moderating effect of gender on the relationship between physical activity and physical self-efficacy. This finding aligns with the results of previous investigations ([13]). The reasons for this may be partly related to physiological factors. [37] ([37]) found that genetic factors have an average effect of 0.995 on slow-twitch muscle fiber in males and 0.922 in females ([36]). Individuals with a predominance of slow-twitch muscle fibers tend to perform better in endurance activities ([28]). Compared to females, males’ innate physical advantages serve as a physiological prerequisite for their generally higher levels of physical activity (PA) ([57]). Particularly after entering puberty, hormonal changes occur, along with an increase in body fat content, leading to lower participation in high-intensity PAs, such as basketball and running, among females ([24]). whereas males are more likely to engage in these activities and experience higher levels of generalized self-efficacy (GSE), indicating a higher level of participation and a preference for such sports ([38]). The previous literature has also examined the relationship between gender, gender roles, and sports participation among college students, with data showing that male college students are more inclined to participate in sports and maintain a higher proportion of regular participation compared to female college students. On the other hand, societal expectations may also play a role. Studies suggest that the differences in PA levels between male and female college students could be due to differing societal expectations of gender roles. Society generally expects females to embody traits such as “intelligence” and “grace”, while expectations for males revolve around “enthusiasm” and “courage”, leading to greater social support for males engaging in PA ([41]). This discrepancy in expectations may diminish females’ motivation to participate in PA, resulting in significantly lower PA levels among females compared to males. Additionally, unique physical and psychological characteristics of female college students may contribute to this discrepancy. Physiological differences inherently result in females having lower physical strength compared to males. Furthermore, the process of socialization may lead females to focus more on social evaluations during physical activities ([44]), while males are more concerned with the performance itself. This may lead females to have a more one-sided and inaccurate self-assessment of their athletic abilities ([31]), potentially attributing failures to their physical capabilities and reflecting on perceived physical decline, resulting in lower levels of self-efficacy and a higher propensity for emotional dysregulation and low sense of accomplishment, which are symptoms of burnout. The increase in male PA is also influenced by cultural norms that encourage male participation in fitness activities. A study utilizing participatory observation and in-depth interviews revealed that the self-presentation and construction of masculinity are achieved through an embodied process of individual experience and social interaction. Fitness provides an important venue for males to seek individualized self-expression and identity within social and cultural norms. Through fitness, males shape and showcase a variety of intrinsic qualities ([77]). Moreover, the lower physical self-efficacy scores among females compared to males may be due to a stronger focus on body image and a higher level of body anxiety among females ([43]). Consequently, the impact of physical activity on physical self-efficacy appears to be gender dependent, substantiating research Hypothesis H5. Numerous studies have investigated the moderating role of gender in the links between physical exercise and self-efficacy, personality traits, and social interactions. For example, Chen Zhangyuan’s cross-sectional study examined the effect of physical exercise on college students’ subjective well-being and identified a moderating role of gender ([14]). The findings of this study align with these previous observations. However, this study is not without limitations, which should be addressed in future research. Firstly, the sample was limited to students from the Zhuhai Campus of Beijing Normal University. Future research could enhance the generalizability of the findings by expanding the sampling scope to include multiple campuses or regions. Second, this study explored only the direct mechanisms among physical activity, psychological resilience, and physical self-efficacy and neglected potential third variables that may influence physical self-efficacy. Future research should consider incorporating additional variables to elucidate the complex pathways through which physical activity impacts physical self-efficacy. Once again, this study employed an anonymous and voluntary online questionnaire method, strictly adhering to ethical standards and fully protecting participants’ privacy. However, there are some challenges. For instance, some participants may be inclined to provide less than truthful or slightly exaggerated responses due to the perceived anonymity of the survey, which could potentially have an impact on the research. Additionally, self-reported scales are susceptible to personal subjective factors, which may lead to inaccuracies in the results. Finally, the cross-sectional design of this study limits our ability to infer causality. Although we found that physical activity partially mediates the effect on physical self-efficacy through psychological resilience, the data were collected at a single time point, precluding the determination of the temporal sequence between variables. Consequently, the findings only indicate a statistical association between variables and do not allow for the inference of causation. Future research could employ longitudinal studies to analyze the dynamic relationships between physical activity, psychological resilience, and self-efficacy for physical activity, or design intervention experiments to validate the persistence of mediating effects. Moreover, since psychological resilience only accounted for a portion of the effect, this suggests the possible presence of other unmeasured mediating variables.

## 5. Conclusions

The present study revealed a significant positive correlation between physical activity, psychological resilience, and physical self-efficacy. Regression analysis revealed that physical activity significantly predicted psychological resilience and physical self-efficacy, accounting for 4.5% and 17.50% of the variance in these outcome variables, respectively. Additionally, psychological resilience partially mediated the relationship between physical activity and physical self-efficacy, with this mediating effect accounting for 20.74% of the total effect. With respect to gender disparities, an analysis of variance revealed significant differences in physical activity and physical self-efficacy between genders. Further investigation of the moderating role of gender suggested that gender is a pivotal moderating factor in the association between physical activity and physical self-efficacy, with males demonstrating higher levels of both physical activity and physical self-efficacy compared to females. Finally, the cross-sectional design of this study precludes the determination of causality. Future research could employ longitudinal or experimental designs to validate the causal relationships between physical activity, psychological resilience, and physical self-efficacy. Moreover, since psychological resilience only accounted for a portion of the effect, this suggests the possibility of other unmeasured mediating variables. Future studies could further investigate these potential mediators.

## Figures and Tables

**Figure 1 behavsci-15-00541-f001:**
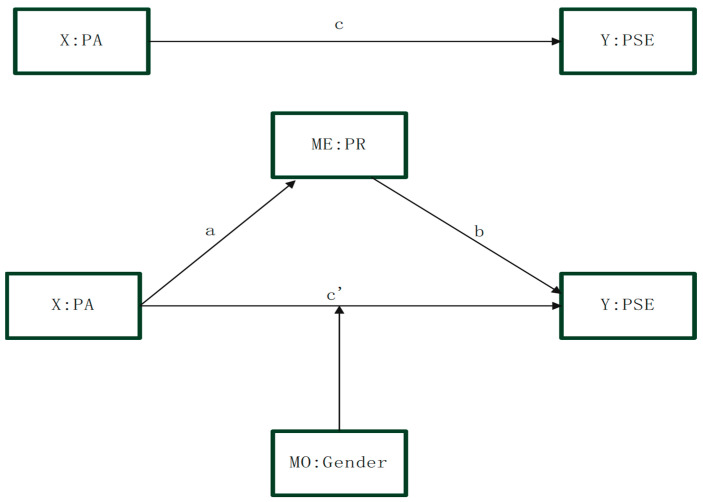
Diagram of the research hypothesis model.

**Figure 2 behavsci-15-00541-f002:**
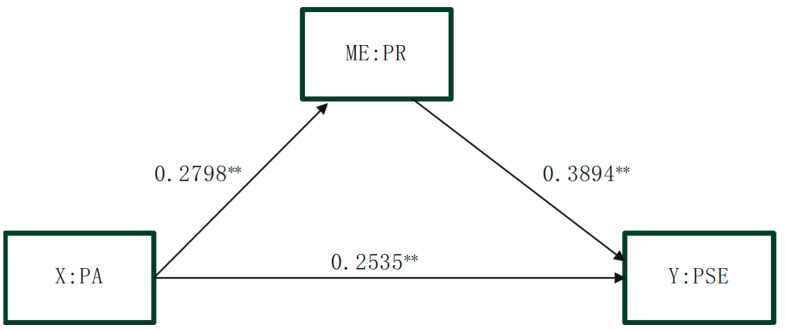
Mediating effect of psychological resilience between physical activity and physical self-efficacy. Note: The path coefficient is expressed as a standardized value (β); ** *p* < 0.01.

**Table 1 behavsci-15-00541-t001:** Summary of data collection.

	Male	Female	Total/Proportion
Freshman	77	163	240/65.0%
Sophomore	28	39	67/18.2%
Junior	24	28	52/14.1%
Senior	4	0	4/1.1%
First-year graduate student	5	1	6/1.6%
Total/Proportion	138/37.1%	231/62.9%	369

**Table 2 behavsci-15-00541-t002:** Results of confirmatory factor analysis for the subscales and the overall scale.

Variable	x2/df	df	RMSEA	GFI	AGFI	NFI	RFI	IFI	TLI	Standardized RMR	Cronbach’s Alpha	AVE	CR
PSE	3.6234	35	0.08	0.93	0.89	0.92	0.90	0.94	0.92	0.0464	0.895	0.50	0.91
PR	2.7792	275	0.06	0.85	0.82	0.85	0.83	0.90	0.89	0.0481	0.939	0.50	0.95

Note: PR: psychological resilience; PSE: physical self-efficacy.

**Table 3 behavsci-15-00541-t003:** Descriptive statistics of the variables by gender for college students.

Variable	Overall	Male (n = 138)	Female (n = 231)	t
PA (M ± SD)	3.4276 ± 0.27797	3.5178 ± 0.29463 **	3.3742 ± 0.25349	−4.940
WHVPA (M ± SD)	1.6491 ± 0.92822	1.8860 ± 0.88967 **	1.4916 ± 0.92123	−4.066
WMVPA (M ± SD)	2.1076 ± 0.51916	2.1740 ± 0.58481 **	2.0679 ± 0.47254	−1.808
WWT (M ± SD)	2.2202 ± 0.51797	2.2170 ± 0.55319	2.2221 ± 0.49698	−0.091
WST (M ± SD)	2.6412 ± 0.19143	2.6239 ± 0.20703	2.6515 ± 0.18115	1.342
PSE (M ± SD)	33.11 ± 9.355	35.43 ± 9.139 **	31.73 ± 9.225	−3.748
PR (M ± SD)	86.57 ± 13.884	88.51 ± 13.159 **	85.41 ± 14.201	−2.085

PR: Psychological resilience; PA: Physical activity; PSE: Physical self-efficacy; WHVPA: Weekly high-intensity physical activity time; WMVPA: Weekly moderate-intensity physical activity time; WWT: Weekly walking time; WST: Weekly sedentary time. ** *p* < 0.01.

**Table 4 behavsci-15-00541-t004:** Pearson bivariate bilateral correlation coefficients for the examined relationships.

	Skewness (Se)	Kurtosis (Se)	WHVPA	WMVPA	WWT	WST	PA	PSE	PR
WHVPA	−0.854 (0.127)	−0.688 (0.253)	1						
WMVPA	−0.940 (0.127)	0.445 (0.253)	0.201 **	1					
WWT	−0.995 (0.127)	0.742 (0.253)	0.110 *	0.127 *	1				
WST	−0.658 (0.127)	2.611 (0.253)	0.015	0.009	0.026	1			
PA	−0.218 (0.127)	−0.127 (0.254)	0.629 **	0.540 **	0.502 **	0.115 *	1		
PSE	0.147 (0.127)	−0.397 (0.253)	0.339 **	0.226 **	0.068	−0.124 *	0.386 **	1	
PR	0.338 (0.127)	−0.016 (0.253)	0.300 **	0.171 **	0.088	0.016	0.289 **	0.471 **	1

Note: ** indicates a significant correlation at the 0.01 level (two-tailed), * indicates a significant correlation at the 0.05 level (two-tailed). PR: Psychological resilience; PA: Physical activity; PSE: Physical self-efficacy; WHVPA: Weekly high-intensity physical activity time; WMVPA: Weekly moderate-intensity physical activity time; WWT: Weekly walking time; WST: Weekly sedentary time.

**Table 5 behavsci-15-00541-t005:** Regression analysis indices for each equation.

	Fitting Condition		ANOVA		Coefficient						
	R^2^	R^2^adj	F	*p*	B	SE	β	t	*p*	q	Significant
Equation (1)	0.149	0.147	64.149	<0.001	12.985	1.621	0.386	8.009	<0.001	0.0028	TRUE
Equation (2)	0.083	0.081	33.344	<0.001	0.006	0.001	0.289	5.774	<0.001	0.0056	TRUE
Equation (3)	0.222	0.220	104.690	<0.001	0.317	0.031	0.471	10.232	<0.001	0.0083	TRUE
Equation (4)	0.292	0.288	75.165	<0.001	0.265	0.031	PR 0.394	8.572	<0.001	0.0111	TRUE
					9.153	1.547	PA 0.272	5.916	<0.001	0.0139	TRUE
Equation (5)	0.158	0.149	17.111	<0.001	3.088	0.497	WHVPA 0.306	6.217	<0.001	0.0167	TRUE
					2.953	0.890	WMVPA 0.164	3.319	<0.001	0.0194	TRUE
					0.299	0.879	WWT 0.017	0.340	0.734	0.0417	FALSE
					−6.369	2.351	WST −0.130	−2.709	0.007	0.0278	TRUE
Equation (6)	0.105	0.095	10.673	0.001	4.078	0.760	WHVPA 0.273	5.366	0.001	0.0250	TRUE
					2.964	1.362	WMVPA 0.111	2.176	0.030	0.0361	TRUE
					1.205	1.346	WWT 0.045	0.895	0.371	0.0444	FALSE
					−1.612	3.598	WST −0.022	−0.448	0.654	0.0472	FALSE
Equation (7)	0.293	0.284	30.143	<0.001	2.020	0.473	WHVPA 0.200	4.268	<0.001	0.0222	TRUE
					2.177	0.822	WMVPA 0.121	2.650	0.008	0.0333	TRUE
					−0.016	0.808	WWT −0.001	−0.020	0.984	0.0500	FALSE
					−5.947	2.157	WST −0.122	−2.756	0.006	0.0306	TRUE
					0.262	0.031	PR 0.389	8.331	<0.001	0.0389	TRUE

Note: R^2^ = coefficient of determination; R^2^adj = adjusted R^2^; β = standardized regression coefficients; B = unstandardized coefficients; SE = standard error; Significance level was set at α = 0.05; PR: Psychological Resilience; PA: Physical Activity; WHVPA: Weekly high-intensity physical activity time; WMVPA: Weekly moderate-intensity physical activity time; WWT: Weekly walking time; WST: Weekly sedentary time.

**Table 6 behavsci-15-00541-t006:** Decomposition of total effects, direct effects, and mediating effects.

	Standardized Effects	Boot SE	Boot CI LL	Boot CI UL	*p*	RES
Total Effect	0.3624	0.0496	0.2649	0.4599	0.0000	
Direct Effect	0.2535	0.0471	0.1608	0.3461	0.0000	69.95%
Mediating Effect of Psychological Resilience	0.1089	0.0238	0.0658	0.1592		30.05%

Note: Independent variable: physical activity; dependent variable: physical self-efficacy; mediating variable: psychological resilience; REE: relative effect size.

**Table 7 behavsci-15-00541-t007:** Decomposition of total effects, direct effects, and mediating effects.

	Standardized Effects	Boot SE	Boot CI LL	Boot CI UL	*p*	RES
Total Effect	0.3123	0.0498	0.2143	0.4103	0.0000	
Direct Effect	0.1965	0.0475	0.1031	0.2899	0.0000	62.91%
Mediating Effect of Psychological Resilience	0.1158	0.0233	0.0720	0.1625		37.09%

Note: Independent variable: high-intensity physical activity; dependent variable: physical self-efficacy; mediating variable: psychological resilience; RES: relative effect size.

**Table 8 behavsci-15-00541-t008:** Decomposition of total effects, direct effects, and mediating effects.

	Standardized Effects	Boot SE	Boot CI LL	Boot CI UL	*p*	RES
Total Effect	0.2158	0.0506	0.1164	0.3152	0.0000	
Direct Effect	0.1447	0.0460	0.0542	0.2352	0.0018	67.05%
Mediating Effect of Psychological Resilience	0.0711	0.0237	0.0262	0.1199		32.95%

Note: Independent variable: moderate-intensity physical activity; dependent variable: physical self-efficacy; mediating variable: psychological resilience; RES: relative effect size.

**Table 9 behavsci-15-00541-t009:** Key goodness-of-fit indicators for the male model.

Indicator Category	Metric Title	Default Model
CMIN	CMIN/DF	1.6432
RMR, GFI	RMR	35.5827
	GFI	0.7073
	AGFI	0.6761
Baseline Comparison	CFI	0.7943
	TLI	0.7830
RMSEA	RMSEA	0.0685
	LO 90	0.0619
	HI 90	0.0750
Simplified Adjustment Indicators	PNFI	0.5754
	PCFI	0.7528
AIC	AIC	1514.8563
ECVI	ECVI	11.0573

**Table 10 behavsci-15-00541-t010:** Key goodness-of-fit indicators for the female model.

Indicator Category	Metric Title	Default Model
CMIN	CMIN/DF	1.8003
RMR, GFI	RMR	39.2491
	GFI	0.7706
	AGFI	0.7462
Baseline Comparison	CFI	0.8581
	TLI	0.8502
RMSEA	RMSEA	0.0590
	LO 90	0.0541
	HI 90	0.0638
Simplified Adjustment Indicators	PNFI	0.6928
	PCFI	0.8132
AIC	AIC	1643.0427
ECVI	ECVI	7.1437

**Table 11 behavsci-15-00541-t011:** Test of the main effects of the gender variable.

IV	DV	Male: M ± SD	Female: M ± SD	DF	MS	F	*p*	R^2^
gender	WHVPA	1.8994 ± 0.87871	1.4916 ± 0.92123	1	14.305	17.440	<0.001	0.045
	WMVPA	2.1833 ± 0.57667	2.0679 ± 0.47254	1	1.145	4.338	0.038	0.012
	WWT	2.2266 ± 0.54358	2.2221 ± 0.49698	1	0.002	0.007	0.936	0.000
	WST	2.6285 ± 0.20074	2.6515 ± 0.18115	1	0.046	1.285	0.258	0.003
	PA	3.5178 ± 0.29463	3.3742 ± 0.25349	1	1.773	24.407	<0.001	0.063
	PR	88.55 ± 13.199	85.41 ± 14.201	1	848.179	4.430	0.036	0.012
	PES	35.36 ± 9.135	31.73 ± 9.225	1	1137.983	13.469	<0.001	0.035

Note: IV: independent variable; DV: dependent variable; DF: degrees of freedom; MS: mean square; PR: psychological resilience; PA: physical activity; PSE: physical self-efficacy; WHVPA: weekly high-intensity physical activity time; WMVPA: weekly moderate-intensity physical activity time; WWT: weekly walking time; WST: weekly sedentary time.

**Table 12 behavsci-15-00541-t012:** Regression analysis by gender.

Gender	R^2^	Adjusted R^2^	F	B	SE	β	t	*p*	Cohen’s f^2^
Male	0.207	0.201	35.146	14.092	2.377	0.454	5.928	<0.001	0.260
Female	0.085	0.081	21.313	10.619	2.300	0.292	4.617	<0.001	0.093

Note: Physical activity is the independent variable, and physical self-efficacy is the dependent variable.

## Data Availability

The data supporting the findings are available within the article.

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
