# Peer review of "The Impact of Physical Exercise on College Students’ Physical Self-Efficacy: The Mediating Role of Psychological Resilience"

_behavsci, 2025, doi:10.3390/bs15040541_

Round 1
Reviewer 1 Report
Comments and Suggestions for Authors
The reviewer will provide the researcher with suggestions for improvement. Researchers are encouraged to review them and incorporate the suggested corrections into their study."
First, I believe the relationship between the variables has not been presented correctly. In addition to explaining the variables, please incorporate findings from the literature to clarify their relationships.
Second, please categorize your research hypotheses. For each hypothesis (H1 to H5), clearly specify the relationships between the variables based on previous research.
Third, the reviewers find that presenting an exploratory factor analysis in the feasibility section is unnecessary. However, please include a specific table for the confirmatory factor analysis. Additionally, physical activity is missing from the confirmatory factor analysis—please consider including it. Furthermore, provide a goodness-of-fit figure for the SRMR at the goodness-of-fit threshold.
Fourth, the results section does not discuss discriminant validity and multicollinearity in the correlation analysis. Please add a detailed discussion on these aspects.
Fifth, organize the discussion according to the research hypotheses. Avoid presenting only partial results; instead, provide a comprehensive discussion of the implications based on previous studies. Clearly explain the similarities and differences between the current findings and prior research.
Author Response
To Reviewer 1
We are profoundly grateful for the reviewer's thorough assessment and precise attention to detail, which have proven indispensable in steering our revisions and elevating the quality of our work to a superior standard.
The manuscript has been rechecked, and the necessary changes have been made in accordance with the reviewers’ suggestions. The responses to all comments have been prepared and attached below.
#1: First, I believe the relationship between the variables has not been presented correctly. In addition to explaining the variables, please incorporate findings from the literature to clarify their relationships.
Thank you for your advice. We updated the Manuscript following your recommendation.We have reassessed the relationships between the variables and provided a more precise and comprehensive explanation within the manuscript. The revisions can be found highlighted throughout pages 1 to 4. We are confident that these alterations enhance the quality of the paper and sharpen the clarity of our argument. We hope that the revised version meets your expectations.
#2: Second, please categorize your research hypotheses. For each hypothesis (H1 to H5), clearly specify the relationships between the variables based on previous research.
Thank you for your advice. We updated the Manuscript following your recommendation. Before presenting our hypotheses, we have elaborated on the findings of previous studies and incorporated theoretical support to clarify the relationships between the variables. You can find this content from line 133 to line 174 on pages 3 to 4. The modified content is highlighted in red font. We hope that the revised version meets your satisfaction.
#3: Third, the reviewers find that presenting an exploratory factor analysis in the feasibility section is unnecessary. However, please include a specific table for the confirmatory factor analysis. Additionally, physical activity is missing from the confirmatory factor analysis—please consider including it. Furthermore, provide a goodness-of-fit figure for the SRMR at the goodness-of-fit threshold.
Thank you for your advice. We updated the Manuscript following your recommendation. In accordance with your suggestion, I have removed the table for exploratory factor analysis and have instead included a detailed confirmatory factor analysis table, which encompasses fit indices for physical activity and standardized RMR indices for each variable. You can locate this in Table 2 on page 6. The modified content is highlighted in red font. We hope that the revised version meets your satisfaction.
#4: Fourth, the results section does not discuss discriminant validity and multicollinearity in the correlation analysis. Please add a detailed discussion on these aspects.
Thank you for your advice. We updated the Manuscript following your recommendation. Thank you for pointing out the need for a discussion on discriminant validity and multicollinearity in the correlation analysis within the results section. I have now included a detailed discussion on these aspects in the revised manuscript. The new content addresses the concerns raised and provides a thorough examination of discriminant validity and multicollinearity, which we believe strengthens the analysis presented. You can find it on page 10, lines 404 to 408. The modified content is highlighted in red font. We hope that the revised version meets your satisfaction.
#5: Fifth, organize the discussion according to the research hypotheses. Avoid presenting only partial results; instead, provide a comprehensive discussion of the implications based on previous studies. Clearly explain the similarities and differences between the current findings and prior research.
Thank you for your advice. We updated the Manuscript following your recommendation. I have revised the discussion section to provide an in-depth analysis of the results for each hypothesis, offering a comprehensive discussion based on previous research. I have also delineated the differences between the current findings and prior studies, thereby enhancing the contextual understanding and significance of our research. You can find these modifications starting from line 562 to line 595 on page 16. The modified content is highlighted in red font. We hope that the revised version meets your satisfaction.
In summary, we wish to express our sincere appreciation to you, the reviewer, for your perceptive and beneficial feedback. Your valuable insights have significantly improved the coherence and robustness of our manuscript, while also steering us in refining our methodology and discussion of the topic. We are thankful for the considerable time and energy you have invested in assisting us in enhancing our research, and we are fortunate to be part of an academic community that is both enlightening and supportive.
Reviewer 2 Report
Comments and Suggestions for Authors
Review for paper work: The Impact of Physical Exercise on College Students’ Physical 2 Self-Efficacy: The Mediating Role of Psychological Resilience
The reviewer will provide the researcher with suggestions for improvement. Researchers are encouraged to review them and incorporate the suggested corrections into their study."
First, I believe the relationship between the variables has not been presented correctly. In addition to explaining the variables, please incorporate findings from the literature to clarify their relationships.
Second, please categorize your research hypotheses. For each hypothesis (H1 to H5), clearly specify the relationships between the variables based on previous research.
Third, the reviewers find that presenting an exploratory factor analysis in the feasibility section is unnecessary. However, please include a specific table for the confirmatory factor analysis. Additionally, physical activity is missing from the confirmatory factor analysis—please consider including it. Furthermore, provide a goodness-of-fit figure for the SRMR at the goodness-of-fit threshold.
Fourth, the results section does not discuss discriminant validity and multicollinearity in the correlation analysis. Please add a detailed discussion on these aspects.
Fifth, organize the discussion according to the research hypotheses. Avoid presenting only partial results; instead, provide a comprehensive discussion of the implications based on previous studies. Clearly explain the similarities and differences between the current findings and prior research.
I have no comment to make. I propose accepting the article in this form.
Author Response
To Reviewer 2
#1: I have no comment to make. I propose accepting the article in this form.
We are truly grateful for the valuable feedback and insights provided by you, the reviewers. Thanks to your constructive suggestions, the manuscript has been significantly strengthened. I'm eager to collaborate with you to facilitate the publication of this manuscript in Behavioral Sciences. The manuscript has undergone a thorough review, and appropriate revisions have been made in accordance with your suggestions. Replies to all comments have been compiled and are included below. Thank you again for your invaluable advice.
Reviewer 3 Report
Comments and Suggestions for Authors
This article has the following issues:
- Sampling and Generalizability
Issue: You drew participants from a single campus without demonstrating representativeness or random selection.
• Recommendation: Expand your sampling frame to include multiple institutions or regions. Provide explicit power analyses or rationales for sample size to substantiate statistical claims. - Measurement Quality (IPAQ and Scales)
Issue: The short IPAQ is prone to self-report errors, and you neither detailed methods to mitigate overreporting/underreporting nor addressed its low KMO value (0.494).
• Recommendation: Employ objective measures (e.g., accelerometers) or at least validate IPAQ scores with more robust strategies. If the KMO remains too low, consider revising or removing problematic items and re-checking factor structure. - Reliability and Validity Across Groups
Issue: You compare male vs. female data without confirming measurement invariance of the scales for each gender.
• Recommendation: Perform multi-group CFAs to ensure that factor loadings, intercepts, and error variances are equivalent. Only then should you interpret gender differences as genuine. - Statistical Assumptions
Issue: There is no mention of tests for normality, homoscedasticity, or outlier influences in your regression/mediation models.
• Recommendation: Report results of normality checks (e.g., Shapiro–Wilk), homoscedasticity, and outlier detection. If assumptions are violated, use appropriate transformations or robust analyses (e.g., robust regression). - Multicollinearity Checks
Issue: Vigorous, moderate, and walking activities can correlate highly, yet you do not provide VIF or tolerance values.
• Recommendation: Check VIF/tolerance. If multicollinearity is severe, consider combining highly correlated physical activity indices or using dimension-reduction techniques before regression. - Mediation Analysis (Partial Mediation)
Issue: You frame your findings in causal language but employed a cross-sectional design that cannot establish temporal precedence.
• Recommendation: Clarify that you observed only statistical associations. If you seek causal assertions, use longitudinal or experimental designs. Explicitly note that the mediating variable (resilience) explains only part of the effect, implying other unmeasured mediators may be relevant. - Moderation by Gender
Issue: The Fisher’s Z test is minimally described, and the effect size for differences in slopes is unclear.
• Recommendation: Show unstandardized slopes for each gender, their standard errors, and effect sizes (e.g., Cohen’s f²) to demonstrate whether differences are substantively meaningful rather than only statistically significant. - Multiple Comparisons and Type I Error
Issue: The paper includes several ANOVAs and regressions without any correction for multiple testing.
• Recommendation: Apply a false discovery rate (FDR) or Bonferroni-type correction, or at least discuss the risk of inflated Type I error rates. - Reporting of Effect Sizes
Issue: Mediation path coefficients (e.g., 0.0004, 0.0015) are unclear in their scale or interpretation. R² values (e.g., 0.045) receive minimal context.
• Recommendation: Specify whether these coefficients are standardized or unstandardized and tie them to meaningful units (e.g., “per 100 MET-min/week”). Contextualize R² values in terms of small, medium, or large effects. - Avoiding Overreach in Conclusions
- Issue: You frequently use wording like “physical activity enhances…” despite your cross-sectional data.
• Recommendation: Replace causal statements with association-based language (“is linked to” or “is associated with”). Suggest follow-up longitudinal or intervention research to validate cause-effect claims.
Author Response
To Reviewer 3
We wish to express our deepest appreciation for the comprehensive evaluation and sharp focus you have brought to the review of our manuscript. Your insights have been instrumental in guiding our revisions and have undoubtedly enhanced the caliber of our work to a higher level.
We have meticulously reviewed the manuscript once more, incorporating the essential modifications based on your invaluable recommendations. The process has been one of growth and refinement, and we believe the manuscript is much improved as a result.
We have also compiled detailed responses to each of your comments, which are appended below. Your feedback has been a catalyst for us to delve deeper into the subject matter and to present our findings with greater clarity and precision.
We are committed to ensuring that our work meets the high standards expected by Behavioral Sciences, and we greatly appreciate your continued support and expertise, which are invaluable to us as we advance in this journey of scholarly inquiry.
This article has the following issues:
#1: Sampling and Generalizability
Issue: You drew participants from a single campus without demonstrating representativeness or random selection.
• Recommendation: Expand your sampling frame to include multiple institutions or regions. Provide explicit power analyses or rationales for sample size to substantiate statistical claims.
Thank you for your advice. We updated the Manuscript following your recommendation. Firstly, our study sample consists of participants from different grades and genders, ensuring the diversity of the sample population. You can find this information on page 5, lines 188 to 195. Secondly, I have mentioned the limitation of the small sample size in the discussion section, suggesting that future research should expand the scope to include multiple campuses or regions to enhance the generalizability of the research findings. You can find this discussion on page 18, lines 671 to 674. The modified content is highlighted in red font. We hope that the revised version meets your satisfaction.
#2: Measurement Quality (IPAQ and Scales)
Issue: The short IPAQ is prone to self-report errors, and you neither detailed methods to mitigate overreporting/underreporting nor addressed its low KMO value (0.494).
• Recommendation: Employ objective measures (e.g., accelerometers) or at least validate IPAQ scores with more robust strategies. If the KMO remains too low, consider revising or removing problematic items and re-checking factor structure.
Thank you for your advice. We updated the Manuscript following your recommendation. The International Physical Activity Questionnaire covers physical activities of varying intensities, as well as walking and sedentary behavior, providing a comprehensive picture of physical activity. However, the IPAQ relies on participants' recall, which may be influenced by recall bias, leading to inaccuracies in the reported levels of physical activity. Therefore, in the discussion section, I addressed the issues that are commonly associated with self-reported data. You can find this content on page 18, lines 679 and 680. The modified content is highlighted in red font. We hope that the revised version meets your satisfaction.
#3: Reliability and Validity Across Groups
Issue: You compare male vs. female data without confirming measurement invariance of the scales for each gender.
• Recommendation: Perform multi-group CFAs to ensure that factor loadings, intercepts, and error variances are equivalent. Only then should you interpret gender differences as genuine.
Thank you for your advice. We updated the Manuscript following your recommendation. Following your recommendation, I have conducted multi-group confirmatory factor analyses (CFAs) to ensure that factor loadings, intercepts, and error variances are equivalent across genders. I have carried out these multi-group CFAs to test for measurement invariance between males and females. You can find this section on lines 476-489 of page 13, as well as in Tables 9 and 10 on pages 13 and 14, respectively. The modified content is highlighted in red font. We hope that the revised version meets your satisfaction.
#4: Statistical Assumptions
Issue: There is no mention of tests for normality, homoscedasticity, or outlier influences in your regression/mediation models.
• Recommendation: Report results of normality checks (e.g., Shapiro–Wilk), homoscedasticity, and outlier detection. If assumptions are violated, use appropriate transformations or robust analyses (e.g., robust regression).
Thank you for your advice. We updated the Manuscript following your recommendation. We used the one-sample Kolmogorov-Smirnov test (KS test) to examine the normality of the data. Additionally, we employed the Bootstrap method (with 5000 resamples) to estimate the standard errors and confidence intervals of regression coefficients for non-normally distributed data, thereby reducing the reliance on the normality assumption. You can find this section on lines 367 to 376 of page 9. The modified content is highlighted in red font. We hope that the revised version meets your satisfaction.
#5: Multicollinearity Checks
Issue: Vigorous, moderate, and walking activities can correlate highly, yet you do not provide VIF or tolerance values.
• Recommendation: Check VIF/tolerance. If multicollinearity is severe, consider combining highly correlated physical activity indices or using dimension-reduction techniques before regression.
Thank you for your advice. We updated the Manuscript following your recommendation. Following your advice, I have thoroughly checked the Variance Inflation Factors (VIF) for these variables. All the VIF values are significantly below the commonly accepted threshold of 7, indicating that multicollinearity is not a significant issue in our regression analysis. The VIF values have now been included in the revised manuscript. You can find this section on lines 394 to 398 of page 10. The modified content is highlighted in red font. We hope that the revised version meets your satisfaction.
#6: Mediation Analysis (Partial Mediation)
Issue: You frame your findings in causal language but employed a cross-sectional design that cannot establish temporal precedence.
• Recommendation: Clarify that you observed only statistical associations. If you seek causal assertions, use longitudinal or experimental designs. Explicitly note that the mediating variable (resilience) explains only part of the effect, implying other unmeasured mediators may be relevant.
Thank you for your advice. We updated the Manuscript following your recommendation. Following your advice, I have revised this section to clarify that the research findings represent statistical associations rather than causal relationships, which is consistent with the cross-sectional nature of the study design. In the discussion section, I have included an explanation to emphasize that although the mediating variable (resilience) accounts for part of the effect, this study does not establish a causal relationship. Additionally, I have highlighted the possibility of other unmeasured mediators and the potential value of future research using longitudinal or experimental designs to investigate causal relationships. You can find this content on lines 670-680 of page 18, and from line 693 to 670. The modified content is highlighted in red font. We hope that the revised version meets your satisfaction.
#7: Moderation by Gender
Issue: The Fisher’s Z test is minimally described, and the effect size for differences in slopes is unclear.
• Recommendation: Show unstandardized slopes for each gender, their standard errors, and effect sizes (e.g., Cohen’s f²) to demonstrate whether differences are substantively meaningful rather than only statistically significant.
Thank you for your advice. We updated the Manuscript following your recommendation. I have expanded the description of the Fisher’s Z test to provide a clearer understanding of its application in our study. In response to your recommendation, I have now included the unstandardized slopes for each gender, along with their standard errors and effect sizes (Cohen’s f²) in the results section. This addition aims to demonstrate whether the differences in slopes are not only statistically significant but also substantively meaningful. You can find this content on lines 527 to 529 on page 15, as well as in Table 12. The modified content is highlighted in red font. We hope that the revised version meets your satisfaction.
#8: Multiple Comparisons and Type I Error
Issue: The paper includes several ANOVAs and regressions without any correction for multiple testing.
• Recommendation: Apply a false discovery rate (FDR) or Bonferroni-type correction, or at least discuss the risk of inflated Type I error rates.
We acknowledge the significance of this issue and, in line with your suggestion, have applied the False Discovery Rate (FDR) correction method to the regression analysis to address the problem of inflation in Type I error rates. This correction has been fully implemented in the regression analysis section of the paper, and the revised results have been presented accordingly. You can find this section on lines 389 to 412 of page 10, as well as in the discussion section on lines 538 to 549 of pages 15 and 16. Thank you for your advice. We updated the Manuscript following your recommendation. We acknowledge the significance of this issue and, in line with your suggestion, have applied the False Discovery Rate (FDR) correction method to the regression analysis to address the problem of inflation in Type I error rates. This correction has been fully implemented in the regression analysis section of the paper, and the revised results have been presented accordingly. You can find this section on lines 389 to 412 of page 10, as well as in the discussion section on lines 538 to 549 of pages 15 and 16. The modified content is highlighted in red font. We hope that the revised version meets your satisfaction.
#9: Reporting of Effect Sizes
Issue: Mediation path coefficients (e.g., 0.0004, 0.0015) are unclear in their scale or interpretation. R² values (e.g., 0.045) receive minimal context.
• Recommendation: Specify whether these coefficients are standardized or unstandardized and tie them to meaningful units (e.g., “per 100 MET-min/week”). Contextualize R² values in terms of small, medium, or large effects.
Thank you for your advice. We updated the Manuscript following your recommendation. Following your suggestion, we have clearly indicated in the text whether the mediation path coefficients are standardized or unstandardized. We have associated the coefficients with meaningful units (e.g., "per 100 MET-min/week") to enhance their interpretability. We have provided context for the R-squared values by categorizing them as small, medium, or large effects to better communicate their significance. You can find this content on lines 370 to 391 of pages 9 and 10, as well as on lines 430 to 451 of page 11. The modified content is highlighted in red font. We hope that the revised version meets your satisfaction.
#10: Avoiding Overreach in Conclusions
Issue: You frequently use wording like “physical activity enhances…” despite your cross-sectional data.
• Recommendation: Replace causal statements with association-based language (“is linked to” or “is associated with”). Suggest follow-up longitudinal or intervention research to validate cause-effect claims.
Thank you for your advice. We updated the Manuscript following your recommendation. In line with your suggestion, we have thoroughly reviewed the full manuscript and replaced instances of causal language with association-based language, such as "is related to" or "is associated with." Moreover, we have added recommendations for future longitudinal or intervention studies to validate any claims of causality. I have already replaced the causal language in all highlighted sentences in the introduction and discussion sections with association-based terms. The modified content is highlighted in red font. We hope that the revised version meets your satisfaction.
In conclusion, we extend our heartfelt gratitude to you, the reviewers, for your thoughtful and constructive feedback. Your insights have not only enhanced the clarity and rigor of our manuscript but have also guided us in refining our approach to the subject matter. We appreciate the time and effort you have dedicated to helping us improve our work, and we are grateful for the opportunity to engage with such a knowledgeable and supportive academic community.
Reviewer 4 Report
Comments and Suggestions for Authors
The article expresses an actually aim being focused on presenting a cross-sectional research design that examines the influence of physical activity on physical self-efficacy by incorporating psychological resilience and gender as additional variables. To sustain this approach, the researchers reveal the mediating role of psychological resilience and the moderate effect of gender, adding both theoretical and practical value. Statistical instruments are used properly to offer specific data interpretations and highlight the strengths of the scientific approach.
Still, there are some explanations that may lean to an assumption than being sustained with strength point by the data and these facts are referring to the influence of cultural norms on male physical activity levels.
I think that the approach can be improved by providing deeper interpretation of gender-related findings.
Author Response
To Reviewer 4
Thank you for your recognition and encouragement of our research. We are immensely grateful for your meticulous review and valuable insights, which have been instrumental in refining our manuscript. Your observations have guided us in making substantial improvements, resulting in a more robust and refined paper.
We have carefully addressed each of your suggestions, and the revisions have been enlightening, enhancing the depth and clarity of our work. Our detailed responses to your feedback are provided below.
Your support and expertise are crucial as we aim to meet the high standards of Behavioral Sciences. We look forward to further engaging with you in our scholarly journey.
#1: The article expresses an actually aim being focused on presenting a cross-sectional research design that examines the influence of physical activity on physical self-efficacy by incorporating psychological resilience and gender as additional variables. To sustain this approach, the researchers reveal the mediating role of psychological resilience and the moderate effect of gender, adding both theoretical and practical value. Statistical instruments are used properly to offer specific data interpretations and highlight the strengths of the scientific approach.
Still, there are some explanations that may lean to an assumption than being sustained with strength point by the data and these facts are referring to the influence of cultural norms on male physical activity levels.
I think that the approach can be improved by providing deeper interpretation of gender-related findings.
Thank you for your advice. We updated the Manuscript following your recommendation. In the discussion section, I have cited the findings of previous related studies to support my arguments. I have included references to factors that influence physical activity levels. You can find this content on page 17, from lines 621 to 664. The modified content is highlighted in red font. We hope that the revised version meets your satisfaction.
Finally, I would like to extend my heartfelt gratitude to you for your meticulous attention and valuable guidance throughout the review process. Your insightful suggestions have significantly enhanced the quality of the manuscript. I am also deeply appreciative of your thorough and constructive feedback. Your dedication and expertise have been instrumental in refining the paper, and I am sincerely grateful for your time and effort.
Round 2
Reviewer 1 Report
Comments and Suggestions for Authors
First, the introduction still does not explain the relationship between the variables within the context of the study. Do not end with a simple description of the variables. Instead, you should establish the need for the study by highlighting the relationships among variables and referencing previous research.
Second, I noticed that the exploratory factor analysis figure was removed from the methods section, but references to the analysis remain in the text. This inconsistency needs to be addressed. Also, the Cronbach’s alpha coefficients are missing from the figure. Please either add the Cronbach’s alpha values to Table 2 or present the AVE and CR values as is typically done.
Third, the AVE values for the correlations should be included in Table 2. In addition, skewness and kurtosis values must be reported. For correlation coefficients that are not statistically significant, please provide an explanation—why is there no correlation? Some theoretical or contextual background is needed.
Fourth, if a modified R-squared was used in the regression analysis, it should be indicated with the proper mathematical notation. Since different researchers may use different scales, standardized coefficients should be used. Make it clear—either in the figure or in the text—that the coefficients are standardized.
Author Response
To Reviewer 1
We are profoundly grateful for the reviewer's thorough assessment and precise attention to detail, which have proven indispensable in steering our revisions and elevating the quality of our work to a superior standard. During this revision process, to further validate the robustness of the research results, we additionally collected 100 valid samples and re-conducted the data analysis. A larger sample size can reduce the impact of random errors, making the research results closer to the true situation of the population, thereby enhancing the reliability of the study.
The manuscript has been rechecked, and the necessary changes have been made in accordance with the reviewers’ suggestions. The responses to all comments have been prepared and attached below.
#1: First, the introduction still does not explain the relationship between the variables within the context of the study. Do not end with a simple description of the variables. Instead, you should establish the need for the study by highlighting the relationships among variables and referencing previous research.
Thank you for your advice. We updated the Manuscript following your recommendation.We have reassessed the relationships between the variables and provided a more precise and comprehensive explanation within the manuscript.
First, Before proposing Hypothesis 1, we first established a theoretical foundation for the potential bidirectional relationship between physical activity and self-efficacy through Bandura’s (1986) self-efficacy theory. Additionally, in our review of the existing literature, we observed a predominant focus on the correlation between physical activity and general self-efficacy, with a relative lack of investigation into the relationship between physical activity and physical self-efficacy. Furthermore, the majority of studies have concentrated on the positive predictive role of self-efficacy, while the exploration of how physical activity influences self-efficacy remains underexplored, representing a gap in the field. Based on the interrelationships among these variables, Hypothesis 1 is proposed to emphasize the necessity of establishing research on the connection between physical activity and physical self-efficacy. This content can be found on pages 3 to 4, lines 146-155.
Second, previous research has demonstrated that physical activity, as a form of motor behavior, is significantly and positively correlated with psychological resilience. However, the relationship between different intensities of exercise and psychological resilience remains unclear. Consequently, it is plausible that varying activity intensities may construct resilience through distinct psycho-physiological pathways. Therefore, Hypothesis H2 is proposed to establish a foundational correlation, and the study also investigates the relationship between different intensities of physical activity and psychological resilience. The necessity of this research is underscored by emphasizing the interrelation of these variables. This content can be found on page 4, lines 163-165.
Third, Before proposing Hypothesis 3, we reinforced the theoretical foundation by explicitly linking physical self-efficacy to psychological resilience through Bandura’s (1997) self-efficacy theory. This theory posits that efficacy beliefs in specific domains, such as physical capabilities, can generalize to broader psychological adaptation, including stress resilience. By elucidating this relationship, the necessity of this research is established. This content can be found on page 4, lines 176-178.
Fourth, Existing research has not yet validated the complex interrelationships among physical activity, psychological resilience, and self-efficacy, particularly whether psychological resilience mediates the impact of physical exercise on physical self-efficacy, within the college student population. The mediating role of psychological resilience has also not been adequately explored. Consequently, Hypothesis 4 is proposed to establish the necessity of this research. This content can be found on page 4, lines 180-186.
Fifth, Existing research has identified gender as a moderator of the relationship between physical activity and general self-efficacy among college students, with a high degree of correlation observed between general self-efficacy and physical self-efficacy. However, there is currently no empirical evidence to support the moderating effect of gender on the relationship between physical activity and physical self-efficacy in this population. This lack of research underpins the necessity of proposing Hypothesis 5 to suggest the need for further investigation. This content can be found on page 4, lines 191-199.
Finally, We hope that the revised version meets your satisfaction. The modified content is highlighted in blue font.
#2: Second, I noticed that the exploratory factor analysis figure was removed from the methods section, but references to the analysis remain in the text. This inconsistency needs to be addressed. Also, the Cronbach’s alpha coefficients are missing from the figure. Please either add the Cronbach’s alpha values to Table 2 or present the AVE and CR values as is typically done.
Thank you for your advice. We updated the Manuscript following your recommendation. Based on your opinion, it is unnecessary to conduct an exploratory factor analysis; therefore, I have removed the table related to the exploratory factor analysis and the corresponding content from the text. Additionally, following your suggestion, I have supplemented the Cronbach's alpha coefficients. The Cronbach’s alpha values for physical self-efficacy and psychological resilience are 0.895 and 0.939, respectively. Furthermore, we have reported the Average Variance Extracted (AVE) and Composite Reliability (CR) values for physical self-efficacy and psychological resilience as (0.4631, 0.8953) and (0.463, 0.948), respectively. A high CR indicates a higher level of internal consistency of the measurement model, suggesting that the indicators effectively measure the same construct and that the measurement results are more reliable. However, since the short form of the International Physical Activity Questionnaire (IPAQ) typically includes several open-ended questions, such as asking respondents about the frequency and duration of different types of physical activities they engaged in over the past week, the answers to these open-ended questions cannot be directly quantified. As it is not a scale itself, it is not possible to calculate statistical indicators based on these measures. This section of content is located on page 7, lines 280-289, and is also presented in Table 2. The modified content is highlighted in blue font. We hope that the revised version meets your satisfaction.
#3: Third, the AVE values for the correlations should be included in Table 2. In addition, skewness and kurtosis values must be reported. For correlation coefficients that are not statistically significant, please provide an explanation—why is there no correlation? Some theoretical or contextual background is needed.
Thank you for your advice. We updated the Manuscript following your recommendation. According to your suggestion, we have added the AVE values, as well as skewness and kurtosis, to Table 2. For correlation coefficients that do not have statistical significance, we have also provided corresponding theoretical explanations. You can find the textual explanation regarding the AVE values, skewness, and kurtosis on page 7, lines 280-289, along with the specific numerical values in Table 2. For the explanation of correlation coefficients without statistical significance, please refer to the section on pages 17, lines 621-634. The modified content is highlighted in blue font. We hope that the revised version meets your satisfaction.
#4: Fourth, if a modified R-squared was used in the regression analysis, it should be indicated with the proper mathematical notation. Since different researchers may use different scales, standardized coefficients should be used. Make it clear—either in the figure or in the text—that the coefficients are standardized.
Thank you for your advice. We updated the Manuscript following your recommendation. Regarding the use of the adjusted R-squared in the regression analysis, I have represented it with appropriate mathematical symbols, which you can find in the formula below Table 5 on page 11.
Secondly, standardized coefficients (βvalues) have been reported in all analyses, and “standardized coefficients” or the β symbol have been clearly marked in the text, tables, and figures. The use of standardized coefficients eliminates scale-dependent unit differences, thereby enabling direct comparison of regression coefficients across variables measured with different metrics. You can find the relevant text on pages 12, lines 464-486, and see this content in Tables 6, 7, 8, and Figure 2. I have also included annotations below these tables and figures. The modified content is highlighted in blue font. We hope that the revised version meets your satisfaction.
In summary, we wish to express our sincere appreciation to you, the reviewer, for your perceptive and beneficial feedback. Your valuable insights have significantly improved the coherence and robustness of our manuscript, while also steering us in refining our methodology and discussion of the topic. We are thankful for the considerable time and energy you have invested in assisting us in enhancing our research, and we are fortunate to be part of an academic community that is both enlightening and supportive.
Reviewer 3 Report
Comments and Suggestions for Authors
From a statistical point of view, the following are still problems:
- Sampling is a big problem for external validity: this study is not generalisable. We cannot consider that the results apply to any other population than the one actually included in the sample.
- No power analysis is done to justify the sample size, so we do not know what the statistical power is, especially with regard to detecting weaker effects: not only is it not well sampled in terms of representativeness, but the sample size is also not statistically justified.
- The standard linear regression with small values in some models raises the issue of significance.
- The small size of the indirect effect is overestimated by the authors from the point of view of practical importance
- The adjusted results invalidate some of the secondary findings from Benjamini-Hochberg
- It is not clear at FDR which are the raw p-values and which are the adjusted q-values in the results
- AIC and ECVI could be extended with tests to determine the difference between the models involved.
- There is a lack of longitudinal support for the mediation model.
Author Response
To Reviewer 3
We extend our profound gratitude for the thorough assessment and the keen attention to detail you have provided in reviewing our manuscript. Your perspectives have been pivotal in steering our revisions, undoubtedly elevating the quality of our work to a superior standard.
With meticulous care, we have re-evaluated the manuscript, integrating crucial amendments based on your insightful suggestions. This process of growth and enhancement has led us to believe that the manuscript has significantly progressed.
Furthermore, we have prepared detailed responses to each of your comments, which are included below. Your feedback has inspired us to explore the subject matter more deeply and to articulate our findings with enhanced clarity and precision.
We remain dedicated to upholding the stringent standards of Behavioral Sciences, and we deeply value your ongoing support and expertise, which are indispensable as we continue our scholarly pursuits.
From a statistical point of view, the following are still problems:
#1: Sampling is a big problem for external validity: this study is not generalisable. We cannot consider that the results apply to any other population than the one actually included in the sample.
Thank you for your advice. We updated the Manuscript following your recommendation.
First, during this revision process, to further validate the robustness and generalizability of our research findings, we additionally collected 100 valid samples (increasing the total sample size from 283 to 369) and conducted a complete re-analysis of the data. The results show that the analysis of the newly added samples is highly consistent with the original results, further confirming the stability and reliability of our research findings. This not only enhances our confidence in the research conclusions but also provides strong support for the generalizability of the study.
Second, our study sample consists of participants from different grades and genders, ensuring the diversity of the sample population. By including participants from different grades and genders, the research results can better reflect the diversity of the overall population, thereby enhancing the universality and applicability of the study. You can find this information on page 5, lines 213 to 214.The modified content is highlighted in blue font.
Third, I mentioned the limitation of the small sample size in the discussion section. Future research needs to be expanded to include multiple campuses or regions to enhance the generalizability of the findings. You can find this content on page 19, lines 740 to 743. The modified content is highlighted in blue font. We hope that the revised version meets your satisfaction.
#2: No power analysis is done to justify the sample size, so we do not know what the statistical power is, especially with regard to detecting weaker effects: not only is it not well sampled in terms of representativeness, but the sample size is also not statistically justified.
Thank you for your advice. We updated the Manuscript following your recommendation.
Firstly, during this revision process, to further validate the robustness and generalizability of our research findings, we additionally collected 100 valid samples (increasing the total sample size from 283 to 369) and conducted a complete re-analysis of the data. The results show that the analysis of the newly added samples is highly consistent with the original results, further confirming the stability and reliability of our research findings. This not only enhances our confidence in the research conclusions but also provides strong support for the generalizability of the study.
Second,we have conducted a detailed power analysis using G*Power 3.1 software, and the results confirm that the sample size is statistically justified under the current study design. Specifically, we employed an F-test (linear multiple regression: fixed model, R²deviation from zero) with a small-to-medium effect size (f²= 0.05, Cohen’s standard), a significance level (α= 0.05), and six predictors. With a total sample size of N = 369, the analysis yielded a noncentrality parameter λ = 18.45, a critical F-value = 2.124, and a statistical power (1–β) of 0.913 (91.3%). This result significantly exceeds the commonly recommended threshold of 80% power, demonstrating that our sample size is not only sufficient for detecting medium effects but also exhibits strong statistical sensitivity for weaker effects (e.g., f²= 0.05). You can find this content on page 7, lines 315 to 319. The modified content is highlighted in blue font. We hope that the revised version meets your satisfaction.
#3: The standard linear regression with small values in some models raises the issue of significance.
Thank you for your advice. We updated the Manuscript following your recommendation. In our regression analysis, we reported standardized coefficients (β values) to better compare the effect strengths of the same variables across different studies. You can find them in Table 5 on page 11. Additionally, there is an issue of small standard regression values in the regression analysis. According to Cohen (1988) in "Statistical Power Analysis for the Behavioral Sciences" (pages 284-288), these are classified as "small effects," but they possess both theoretical rationality and practical significance. Cohen noted that due to the complexity of human behavior, small effects are common in behavioral research. Therefore, I provided an explanation in the discussion section, which you can find in lines 582-587 on page 16. The modified content is highlighted in blue font. We hope that the revised version meets your satisfaction.
#4: The small size of the indirect effect is overestimated by the authors from the point of view of practical importance
Thank you for your advice. We updated the Manuscript following your recommendation. In addition, in the previous mediation analysis, we reported non-standardized coefficients. After revision, we now uniformly report standardized coefficients (β values) and clearly mark them as "standardized coefficient" or with the β symbol. The purpose of using standardized coefficients is to eliminate differences between different scale units, allowing for direct comparison of regression coefficients measured by different indicators. The relevant text can be found in lines 464-486 on page 12, as well as in Tables 6, 7, and 8 and Figure 2 spanning pages 12 to 13.
Ultimately, the results of the mediation analysis indicate that the mediating role of psychological resilience accounts for only 24.42% of the total effect, suggesting its relatively limited impact. In light of this, we have thoroughly discussed this finding in the discussion section, pointing out that psychological resilience constitutes only a part of the overall influence, which implies the possible existence of other unmeasured mediating variables. For further details, please refer to lines 749-760 on pages 19-20. The modified content is highlighted in red font. We hope that the revised version meets your satisfaction.
#5:The adjusted results invalidate some of the secondary findings from Benjamini-Hochberg
Thank you for your advice. We updated the Manuscript following your recommendation. As stated in our manuscript, the Benjamini-Hochberg (BH) procedure indeed resulted in the loss of significance for some initially significant effects, particularly for variables with smaller effect sizes (such as the impact of moderate-intensity physical activity on psychological resilience). This change may be attributed to two main factors: Firstly, as the number of tests increases, the correction applied by the Benjamini-Hochberg procedure may become more stringent, which could result in some initially significant p-values failing to meet the adjusted significance threshold. Secondly, in cases of small sample size or low statistical power, the BH procedure might increase the false negative rate due to insufficient test efficiency, potentially leading to the omission of true effects. Both scenarios could cause initial significant findings to lose their significance after multiple comparison corrections. But the core research results (H1-H4) remain stable, supporting their robustness. You can find this section of content on page 16, lines 588-599. The modified content is highlighted in red font. We hope that the revised version meets your satisfaction.
#6: It is not clear at FDR which are the raw p-values and which are the adjusted q-values in the results
Thank you for your advice. We updated the Manuscript following your recommendation. According to your suggestion, we have presented the original p-value and the adjusted q-value, which you can find in Table 5 on page 11. The modified content is highlighted in blue font. We hope that the revised version meets your satisfaction.
#7: AIC and ECVI could be extended with tests to determine the difference between the models involved.
Thank you for your advice. We updated the Manuscript following your recommendation. In addition, we have calculated the AIC and ECVI values for the male and female models, respectively. The specific values are as follows: the AIC for the male model is 1643.0427, and the ECVI is 7.1437; the AIC for the female model is 1514.8563, and the ECVI is 11.0573. Besides AIC and ECVI, we have also compared other key fit indices, such as CMIN/DF, RMR, GFI, AGFI, CFI, TLI, and RMSEA. The results indicate that the female model outperforms the male model on multiple indices, particularly in terms of CFI, TLI, and RMSEA. To further confirm the differences between the models, we conducted a likelihood ratio test, assuming that the female model is an extension of the male model. We calculated the chi-square difference and the degree of freedom difference, and compared the chi-square difference with the critical value. The results show a significant difference between the two models. Although the female model has a higher AIC value, its ECVI value is lower, and it performs better on other fit indices, especially in terms of CFI, TLI, and RMSEA. The statistical test results suggest that there is a significant difference between the two models, which may reflect the different influences of gender on certain psychological or behavioral characteristics.You can find this content in lines 509-524 on pages 13-14, as well as in Tables 9 and 10. The modified content is highlighted in blue font. We hope that the revised version meets your satisfaction.
#8: There is a lack of longitudinal support for the mediation model.
Thank you for your advice. We updated the Manuscript following your recommendation. We sincerely appreciate the important issue raised by the reviewers regarding the longitudinal support for our mediation model. We acknowledge that our cross-sectional design is unable to demonstrate the long-term stability of the mediating effects. Although our study cannot establish a temporal sequence, the proposed mediation pathway is consistent with established theories such as Bandura's self-efficacy theory and the resilience framework. Furthermore, our model is based on prior theory (such as Bandura, 1997), and we believe it provides a crucial first step in understanding these relationships among Chinese college students. In response to the suggestion, we have explicitly discussed this limitation in the manuscript and proposed longitudinal tracking as a key direction for future research. You can find this section of the content on lines 659-668 of page 18. The modified content is highlighted in blue font. We hope these revisions emphasize the practical significance of our current findings.
In closing, I wish to express my profound appreciation for your careful scrutiny and invaluable guidance during the entire review process. Your thoughtful recommendations have markedly improved the manuscript’s quality. I am also deeply thankful for your comprehensive and constructive feedback. Your commitment and expertise have played a crucial role in refining the paper, and I am genuinely grateful for the time and effort you have invested.
Round 3
Reviewer 1 Report
Comments and Suggestions for Authors
The reviewers appreciate your efforts in revising the paper. However, we would like to point out a few minor issues for your attention and correction.
First, the confirmatory factor analysis in Table 2 is missing the Physical Activitydp variable. As this is an important measure, please make sure to include it.
Second, skewness and kurtosis values are currently shown in Table 2. However, these should be presented in the correlation table instead. Please move them to Table 4. Also, ensure that skewness and kurtosis are reported for all relevant variables, including WHVPA, WMVPA, WWT, WST, PA, PSE, and PR. These values are necessary to assess normality.
Third, in the correlation table (Table 3), the correlation between PSE and PSE, and between PR and PR, is listed as .680. Since these are autocorrelations, the value should be 1.0. Please double-check and revise these results.
Author Response
To Reviewer 1
We express our deep appreciation for the comprehensive evaluation and the meticulous scrutiny you have applied to our manuscript. Your insights have been crucial in guiding our revisions, significantly enhancing the quality of our work to a higher level.
With careful attention to detail, we have re-assessed the manuscript, incorporating essential revisions based on your valuable suggestions. This journey of refinement and improvement has convinced us that the manuscript has made substantial progress.
In addition, we have crafted detailed responses to each of your comments, which are enclosed herein. Your feedback has motivated us to delve deeper into the subject matter and to present our findings with greater clarity and precision.
We remain committed to maintaining the rigorous standards of Behavioral Sciences, and we highly esteem your continued support and expertise, which are invaluable as we advance in our academic endeavors.
#1: First, the confirmatory factor analysis in Table 2 is missing the Physical Activitydp variable. As this is an important measure, please make sure to include it.
Thank you very much for your serious and responsible attitude towards our manuscript. Before this, we have also been paying attention to this issue, and I would like to explain it to you. In our research, we used the "International Physical Activity Questionnaire - Short Form" (IPAQ-SF) to assess physical activity. This is a widely recognized and validated tool for effectively evaluating an individual's level of physical activity. It is not a scale, but mainly collects some data. The IPAQ-SF consists of open-ended questions, such as the frequency and duration of various types of physical activity performed in the past week. These open-ended responses have qualitative characteristics and cannot be directly quantified, therefore traditional statistical indicators (such as CFA fit indices) are not applicable to this part of the data. It is important to note that the IPAQ-SF, as a questionnaire, is not a scale. It aims to collect detailed information about individuals' physical activity habits, providing a data foundation for subsequent analysis. You can find the text content of this section on page 7, lines 277 to 286. The revised content is highlighted in red font. We hope that the revised version will satisfy you. Finally, in previous studies, many have also used the International Physical Activity Questionnaire, and in their research, they did not perform confirmatory factor analysis. I have attached some references that used the International Physical Activity Questionnaire for your reference.
References:
Dunston, E. R., Messina, E. S., Coelho, A. J., Chriest, S. N., Waldrip, M. P., Vahk, A., & Taylor, K. (2022). Physical activity is associated with grit and resilience in college students: Is intensity the key to success?. Journal of American College Health, 70(1), 216-222.
Gerber, M., Kalak, N., Lemola, S., Clough, P. J., Pühse, U., Elliot, C., ... & Brand, S. (2012). Adolescents' exercise and physical activity are associated with mental toughness. Mental health and physical activity, 5(1), 35-42.
Demi̇r, A., & Barut, A. İ. (2020). The relationship between university students' psychological resilience and anxiety levels and comparising in terms of physical activity levels gender and academic achievement. Baltic Journal of Health and Physical Activity, 12(6), 8.
#2: Second, skewness and kurtosis values are currently shown in Table 2. However, these should be presented in the correlation table instead. Please move them to Table 4. Also, ensure that skewness and kurtosis are reported for all relevant variables, including WHVPA, WMVPA, WWT, WST, PA, PSE, and PR. These values are necessary to assess normality.
Thank you for your advice. We updated the Manuscript following your recommendation. I have moved the skewness and kurtosis from Table 2 to Table 4 according to your suggestion. Additionally, the skewness and kurtosis of all relevant variables have been reported, including WHVPA, WMVPA, WWT, WST, PA, PSE, and PR. To ensure the validity of the data and results, we performed a logarithmic transformation on the non-normally distributed data. Logarithmic transformation is a commonly used statistical technique that can normalize the data distribution, thereby meeting the prerequisite assumptions for various statistical analyses. By applying this method, we have enhanced the robustness of our research findings, making the results more reliable. You can find this section of content in Table 4 on pages 6 to 10. The revised content is highlighted in red font. We hope that the revised version meets your satisfaction.
#3: Third, in the correlation table (Table 3), the correlation between PSE and PSE, and between PR and PR, is listed as .680. Since these are autocorrelations, the value should be 1.0. Please double-check and revise these results.
Thank you for your advice. We updated the Manuscript following your recommendation. I have adjusted the values. You can find this section of content in Table 4 on pages 6 to 10. The revised content is highlighted in red font. We hope that the revised version meets your satisfaction.
PS: Finally, I would like to take this opportunity to express my sincere gratitude to you. Your passion for scientific research, your responsibility and meticulousness in work, as well as your precise control of details, all deeply impress me. The profound insight and unique perspectives you have demonstrated have provided valuable guidance for my academic research. For every question you raised, I have seriously reviewed and carefully responded, striving to improve the quality of the article to repay your hard work. Through these numerous revisions, my understanding of the article and my data analysis skills have been greatly enhanced. I am truly thankful for the valuable experience and advice you have given me during the review process, as well as for your recognition of our work. I will live up to expectations and continue to work hard on my future academic journey, doing my utmost to write excellent articles and contribute to the scientific research community.